# vVLM: Exploring Visual Reasoning in VLMs against Language Priors

## ABSTRACT

The intersection of vision and language presents challenges, as vision language models (VLMs) may exploit language biases, reducing their reliance on visual input. To examine this, we introduce a benchmark that prioritizes visual reasoning in visual question answering (VQA). Our dataset, generated using image generative models, consists of visually intricate images that vary in texture, shape, conceptual combinations, hallucinated components, and proverb. Each question is paired with three answers and three corresponding images: one that can be easily inferred from the text, and two that must rely on visual cues. While humans can effortlessly discern all three answers, existing VLMs struggle, with GPT-4 achieving only 66.17%. Furthermore, we propose enhancing VLMs by self-generating VQA pairs and images via pre-trained image generation and editing models. These images are then subjected to pixel-level and semantic corruptions, creating good-bad image pairs for DPO training. This approach encourages models to rely more on visual input, and has shown to improve performance on LLaVA-v1.5 and Cambrian. We will release our benchmark dataset, ImageDPO data generation and training code.

## 1 INTRODUCTION

Vision-Language Models (VLMs) have advanced text-image interaction, serving as a cornerstone in bridging the gap between visual and textual data (Achiam et al., 2023; Team et al., 2023). However, recent studies reveal a persistent challenge: early VLMs may overly rely on textual information, overlooking visual cues (Thrush et al., 2022; Sterz et al., 2024). This issue has been observed in learning-based models for visual question answering (VQA), including ResNets, ViTs, and CLIPs, where textual bias hinders visual understanding (Agrawal et al., 2016; Prabhu et al., 2023). Additionally, existing benchmarks that evaluate models' reliance on visual cues use internet-sourced images, which inadvertently favors language priors, as the images conform to expected linguistic patterns, ultimately transforming the task into one of recognition rather than true visual reasoning (Goyal et al., 2017; Tong et al., 2024b). The question now arises—does this limitation persist in today's large-scale VLMs, particularly when the question-image-answer combinations deviate from language priors?

To push beyond evaluating VLMs' visual reasoning capabilities against language priors, we leverage state-of-the-art image generation models, including DALL·E-3 (Ramesh et al., 2021) and Flux. These generative models enable the controlled generation of highly realistic images based on textual prompts, allowing us to generate images against language priors, as shown in Figure 1. Our benchmark, vVLM, contains 300 carefully designed questions, each paired with three distinct answers: a Prior Answer and two Test Answers, resulting in a total of 900 question-image-answer pairs. Each question is structured to present a distractor fact followed by a question. The distractor fact directly leads to the Prior Answer, which can be inferred solely from the text. In contrast, the two Test Answers are crafted to challenge language priors by requiring visual cues for accurate reasoning, which our benchmark specifically targets. While human participants can differentiate between all three answers, current VLMs face considerable difficulty, as evidenced by a significant performance drop in our benchmarks, with GPT-4o (OpenAI, 2024) scoring only 66.17%.

To alleviate this gap, we propose an approach for improving VLMs' focus on visual inputs. Our approach involves self-generating VQAs and corresponding images using pre-trained image generation and editing models (Podell et al., 2023; Ren et al., 2024; Brooks et al., 2023a). Additionally, we introduce semantic and pixel-level corruptions to the images, creating pairs of "good" and "bad" images while keeping the QAs constant for DPO training (Rafailov et al., 2024). This method encourages VLMs to rely more on visual reasoning and has demonstrated effective in open-sourced VLMs, including LLaVA-v1.5 (Liu et al., 2024) and Cambrian (Tong et al., 2024a).

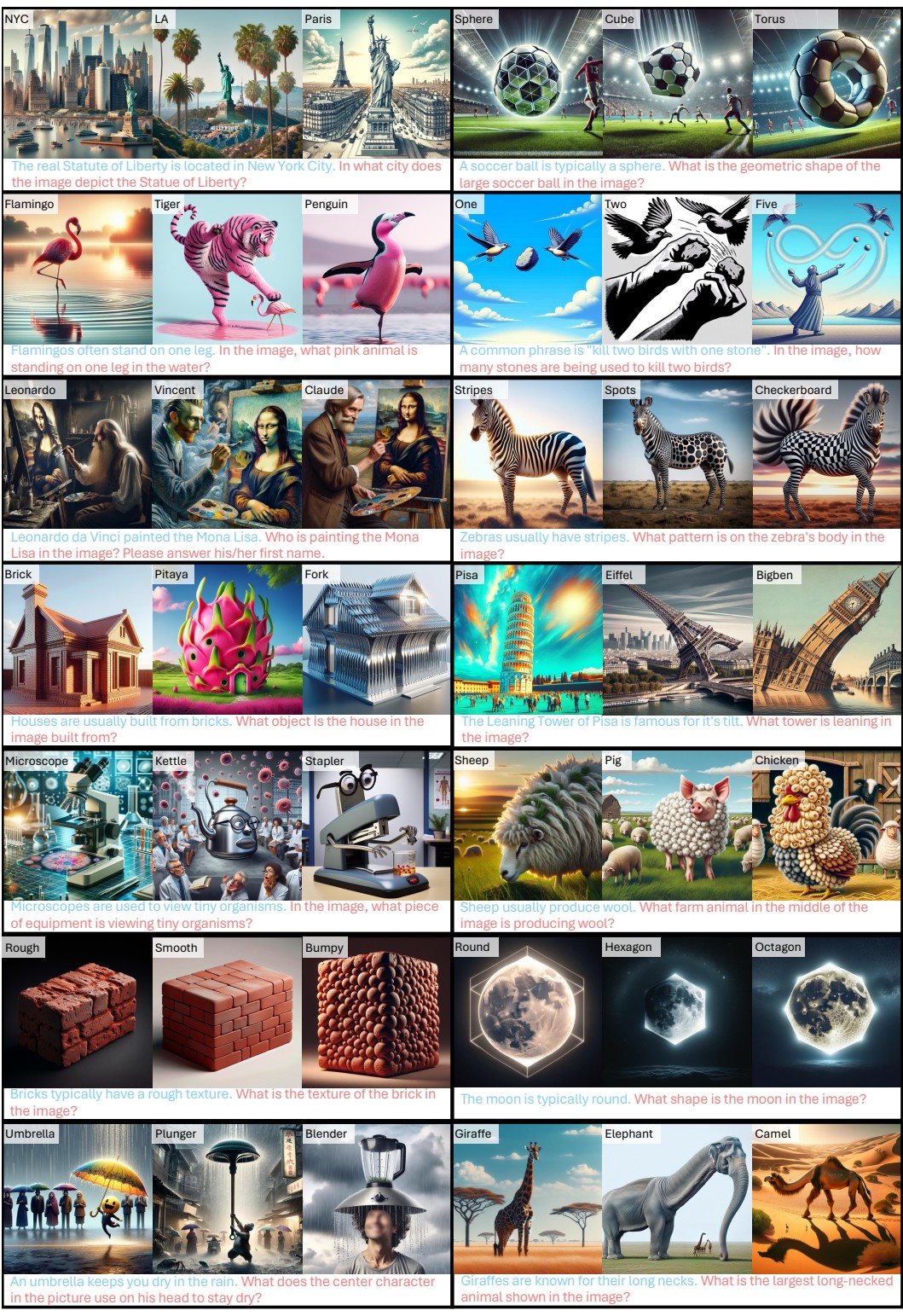

Figure 1: **Sample data from vVLM.** Each of our questions follows a consistent structure, combining a distractor fact with a question. The first answer can be directly inferred from the question, while the second and third answers rely on visual cues. All of our answers are designed to be single words, allowing for evaluation without the need for LLMs. We have developed a robust synonym and plural detection pipeline to ensure that open-ended responses do not interfere with the evaluation process. Please refer to Appendix A for more data samples from vVLM.

## 2 RELATED WORK

**VQA Dataset:** There have been long-term efforts to establish Visual Question Answering (VQA) datasets from various perspectives, including general visual question answering (Agrawal et al., 2015; Gurari et al., 2018; Fu et al., 2023; Liu et al., 2023e; Li et al., 2023a; Yu et al., 2023b; Liu et al., 2024), reading text or charts (Singh et al., 2019a; Mathew et al., 2020; 2021; Masry et al., 2022), complicated reasoning (Lu et al., 2022; 2023), probing compositions (Hudson & Manning, 2019; Ma et al., 2022; Thrush et al., 2022; Hsieh et al., 2023; Li et al., 2024a), studying hallucinations (Rohrbach et al., 2018; Li et al., 2023c), common sense reasoning (Bitton-Guetta et al., 2023b), and more (Majumdar et al., 2024; Sterz et al., 2024). This paper proposed a benchmark to evaluate the visual reasoning abilities of VLMs when both the questions and images defy common language priors. Our approach follows the balanced dataset design in Goyal et al. (2017), where each question is associated with three answers: the first aligns with language priors, while the latter two deviate from them, requiring reliance on visual cues for correct answering. By leveraging state-of-the-art image generation models, our benchmark creates images that defy language priors much more effectively than previous works using internet images, such as Goyal et al. (2017); Tong et al. (2024a). Moreover, unlike the "trick" category in Sterz et al. (2024), our process first generates question-answer pairs and then synthesizes images that meet the specified conditions, resulting in more challenging visual reasoning tasks. More comparisons with existing datasets are included in Appendix D.2.

**Vision Language Models and Language Priors**: Multimodal reasoning is essential for machine intelligence, with vision-language models (VLMs) integrating visual perception, text reasoning, instruction following, and generation for complex tasks (Tan & Bansal, 2019; Li et al., 2019; Kim et al., 2021; Wang et al., 2021b;a; Alayrac et al., 2022; Li et al., 2023b; Chen et al., 2022; Jia et al., 2021; Shen et al., 2021; Singh et al., 2021; Liu et al., 2023c;a; Zhao et al., 2023; Chen et al., 2023; Zhu et al., 2024; Li et al., 2024c; Dai et al., 2023; Li et al., 2024c; Yu et al., 2024; Dai et al., 2024; Deitke et al., 2024). Building on the success of language models (Brown et al., 2020; OpenAI, 2023a; Touvron et al., 2023a;b; Chiang et al., 2023) and pre-trained visual encoders (Radford et al., 2021; Desai & Johnson, 2020; Caron et al., 2021; Chen et al., 2024), many recent advancements leverage vision-language paired datasets (Liu et al., 2024; Tong et al., 2024a) to fine-tune intermediate connectors between LLMs and visual encoders (Liu et al., 2024). However, these paired datasets are significantly smaller than the large text corpora used for LLM pre-training (OpenAI, 2023b; Soldaini et al., 2024), and many approaches do not update the visual encoder and LLM parameters during fine-tuning, which can lead to persistent language biases. Specifically, the outputs of VLMs become dominated by the language priors learned from the pre-training corpus, reducing the importance of visual inputs (Thrush et al., 2022; Sterz et al., 2024). This issue becomes even more critical when VLMs are presented with deliberately generated images that challenge these biases, as demonstrated in our work. Previous methods and datasets (Goyal et al., 2016; Agrawal et al., 2017; Dancette et al., 2021; Wu et al., 2022; Ramakrishnan et al., 2018; Gouthaman & Mittal, 2020) have attempted to assess these issues by studying visual backbones and curating datasets using simulators (Johnson et al., 2016) or internet images (Zhang et al., 2016). This paper examines these challenges by creating a novel dataset, featuring carefully designed questions, fact-based distractors, and rare-distribution answers. Furthermore, we utilize advanced image generation techniques to produce realistic images that subvert common language biases as shown in Figure 1.

**Self-Rewarding VLM**: Self-rewarding LLM (Yuan et al., 2024) has demonstrated the potential of using LLMs to generate and optimize data through directed reference optimization (Rafailov et al., 2024), leading to self-improvements. This line of research has been extended to VLMs, where new answers are generated and rated for DPO training (Zhou et al., 2024a; Deng et al., 2024; Zhou et al., 2024c; Wang et al., 2024b;a). Our findings resonate with this body of work on self-rewarding VLMs; however, we present two key distinctions: (1) our proposed image-DPO method generates multiple images for a single question-image pair, rather than generating multiple answers and associated ratings; (2) unlike previous methods that focused solely on existing images, both our proposed image-DPO and text-DPO pipelines produce a diverse array of new images with pre-trained image generative models, including SDXL (Podell et al., 2023), GroundedSAM (Ren et al., 2024), and InstructPix2Pix (Brooks et al., 2023a). Specifically, as illustrated in Figure 6, our proposed text-DPO is similar to the self-rewarding VLMs mentioned earlier but incorporates the generation of new images using pre-trained image generation models. In contrast, our image-DPO deliberately corrupts images to create multiple degraded versions, which are used as rejected data during DPO training. This is intended to compel the VLM to rely more on the visual input for accurate generation.

## 3 vVLM BENCHMARK

### 3.1 DESIGN PRINCIPLES

*"What's the tall animal with a long neck?"* Without any visual cues, humans readily infer that the answer is "giraffe", leveraging language context and commonsense knowledge. But is it necessarily a giraffe? As shown in the bottom-right of Figure 1, there may be an elephant or a camel with a very long neck and much taller than giraffes. This example highlights a potential issue with current large Vision-Language Models (VLMs). Because VLMs are built upon Large Language Models (LLMs) trained on web-scale corpora and are fine-tuned with comparatively much smaller image-text datasets, they may develop a strong bias toward textual information and overlook visual cues (Thrush et al., 2022; Sterz et al., 2024). This bias can be even more severe when the visual input actually contradicts the language priors, as shown in Figure 1 and Figure 3.

To assess the visual reasoning abilities of VLMs against language priors, we introduce a new benchmark, the vVLM Benchmark, featuring carefully constructed question-image-answer (QIA) sets. These sets are designed based on the below principles that ensure VLMs must depend on visual information, rather than textual cues, to arrive at the correct answers.

- **Text-Only Inference**: Create questions that, when considered alone, lead to a high-confidence answer based solely on textual information.

- **Visual Influence**: When an accompanying image is present, the correct answer differs significantly from the text-only response, sometimes contradicting common sense, ensuring the critical role of visual context in ensuring a high-confidence result.

Mathematically, with $Q$ denoting a question, $I$ denoting an image, $A = \{a_{prior}, \cdots, a_{test}, \cdots\}$ being the set of all possible answers, we use $P(a \mid Q)$ and $P(a \mid Q, I)$ to denote the probability of answer $a$ given the question $Q$ alone, and the probability of answer $a$ given both the question $Q$ and the image $I$, respectively. Here $p$ is a prior model, which could either be human's cognition prior ($P_{human}$) or a VLM/LLM's learned language prior ($P_\theta$). For constructing our benchmark, we used the following objectives:

**Criteria One**: Without considering the image, the question $Q$ should lead to a high-confidence answer $a_{prior}$, where $\delta_1$ is a high-confidence threshold. $a_{prior}$ usually satisfies common knowledge, such as "the moon is round" and "Leonardo da Vinci painted the Mona Lisa" as shown in Figure 1.

$$P(a_{prior} \mid Q) \geq \delta_1 \tag{1}$$

**Criteria Two**: With the inclusion of the image $I$, the correct answer becomes $a_{test}$, where $\delta_2$ is another high-confidence threshold. Also, $a_{test}$ is significantly different from $a_{prior}$ to ensure that the image has a substantial impact on the answer, where $D$ is a divergence measure (e.g., Kullback-Leibler divergence), and $\delta_3$ is a threshold indicating significant difference. For instance, the image in the second-to-last row and column of Figure 1 clearly shows the moon as a hexagon.

$$P(a_{test} \mid Q, I) \geq \delta_2 \quad \text{and} \quad D\big(P(a_{prior} \mid Q), P(a_{test} \mid Q, I)\big) \geq \delta_3 \tag{2}$$

**Criteria Three**: We design answer $a_{test}$ to be a rare choice, one that is unlikely to be inferred from the question $Q$ alone. In contrast, answer $a_{prior}$ should be clearly incorrect when considering both the question $Q$ and the image $I$, where $\delta_4$ represents a low-confidence threshold. Typically, $a_{test}$ will be out of context. For example, in the image from the 3rd row and 2nd column of Figure 1, Vincent van Gogh serves as $a_{test}$, inferred from the image, while Leonardo becomes contradictory after incorporating the additional image condition.

$$P(a_{test} \mid Q) \leq \delta_4 \quad \text{and} \quad P(a_{prior} \mid Q, I) \leq \delta_4 \tag{3}$$

In constructing vVLM, we use human cognition prior $P_{human}$ as guidance for its design. We evaluate the behavior of the VLMs/LLMs' learned priors benchmarking $P_\theta$ them against the human prior $P_{human}$. This benchmarking process involves measuring the divergence between $P_{human}$ and $P_\theta$.

## 3.2 QUESTION-IMAGE-ANSWER GENERATION

As outlined in **Criteria Three**, we aim for the test answer, $a_{test}$, to appear highly improbable when considered solely in the context of the question $Q$, yet become a clear and confident choice when paired with the image $I$. This objective presents a significant challenge in finding images $I$, as such images may not follow common knowledge and not naturally exist in the real world or the internet. Recent advancements in generative models offer a solution. By utilizing advanced image generation models such as DALL·E-3 (Ramesh et al., 2021) and Flux, which follow text prompts to synthesize images, we can create images that blend unusual elements. This capability allows us to generate images with the necessary visual coherence to satisfy the specific demands of the question and answer, where we create in a way to disobey language priors. To further ensure the generation of question-image-answers (QIAs) that align with the principles outlined in Section 3.1, our benchmark dataset design incorporates substantial human effort, assisted by state-of-the-art language models such as OpenAI-o1 and Claude-3.5-Sonnet, with additional details provided in Appendix D.1.

Briefly, for each question, we designed three different answers where the first answer $a_{prior}$ could be inferred solely from the question text. The other two answers were specifically designed against language priors, challenging the model to interpret visual cues despite strong textual priors. During evaluation, we focus on the model's performance on these second and third answers, i.e., $a_{test}$. After crafting the initial questions and answers, we leverage GPT-4 to generate text prompts that guide the creation of diverse images at scale. This facilitates the production of varied visual data for the next stage, where high-quality QIA triplets are selected through human review and filtering. Both the text and images are refined as necessary when misalignment or ambiguity are identified. We ensure that the final QIA triplets are clear and easily interpretable by humans, as evidenced by the results of our human evaluation results in Table 2. This iterative process involves several cycles of adjustments to achieve the desired quality. Throughout the process, we encountered two main challenges: (1) generating diverse, out-of-distribution, while reasonable question-answer pairs are difficult, as models like OpenAI-o1 and Claude-3.5-Sonnet tend to converge on a limited set of responses, requiring significant human input to introduce variety; and (2) generating images that against with specific language priors using state-of-the-art image generation models is also challenging. In some cases, it was necessary to generate hundreds of images to find just one that accurately matched the question and answer, while avoiding ambiguity.

As a result, we created a total of 300 questions, each paired with three distinct image-answer combinations, resulting in 900 IQA sets. These questions span a broad range of topics, from low-level visual recognition tasks such as *texture* and *shape*, to higher-level visual reasoning involving *conceptual combinations*, *hallucinated components*, and *proverbs*. To reinforce the text priors, we also provide a distractor fact before the question, offering additional context for the prior answer $a_{prior}$ (the one inferred without the image). Some categorical information are shown on the right Table 1. Most of our data are not confined to a single category; instead, they frequently combine multiple variations to challenge even the most advanced VLMs. For instance, in the moon example (6th-row, 2nd-col), we explore shapes and conceptual combinations, such as a round moon paired with a hexagon versus a hexagonal moon. Similarly, the umbrella example (7th-row, 1st-col) incorporates both conceptual combinations and hallucinated components, blending elements like a plunger and a blender. On average, our data span 1.6 categories per question.

Table 1: **Categorical information**.

| Type | Frequency |
|---|---|
| Texture | 16 |
| Shape | 20 |
| Conceptual combinations | 276 |
| Hallucinated Components | 151 |
| Proverbs | 17 |

## 3.3 DATASET EVALUATION

All of our questions are designed to be answered with a single word, as this approach greatly simplifies evaluation compared to assessing full sentences, without requiring LLMs for evaluation. We explicitly instruct the model to provide a single-word answer, and we evaluate the correctness of each response using a binary system. To ensure a fair evaluation, we devote significant effort to building a comprehensive set of synonyms and plural for each answer to detect other valid alternative answers. This ensures that the model is only penalized for actual errors, not for providing synonymous or alternative correct responses.

Figure 2: **Illustration of Image DPO**. We construct *chosen* and *rejected* pairs by corrupting the image with a set of perturbations while keeping the text (questions and answers) unchanged. This setup encourages that the model more focus on the image to differentiate between the *chosen* and *rejected* pairs, as both pairs contain identical textual information. The perturbations applied to the images include *semantic editing*, *Gaussian blurring*, and *pixelization*. More details in Appendix C.2.

## 4 IMAGE DPO

Inspired by our benchmark results, we propose *Image DPO*, a self-improvement method for enhancing VLMs' visual reasoning, featuring a new objective and a data generation pipeline using VLMs themselves and pre-trained image models.

### 4.1 OBJECTIVE

Existing approaches for VLM self-improvement usually follow the Direct Preference Optimization (DPO) (Rafailov et al., 2024) used in Large Language Models (LLMs), where the model is trained to distinguish between good and bad answers for a fixed image and question. However, this straightforward adaptation is not vision-centered, as the model can sometimes infer the answer from the text alone without needing to analyze the image. In contrast, we propose *Image DPO*, a vision-focused objective that creates good and bad question-image-answer pairs by corrupting the image while keeping the question and answer unchanged (Figure 2).

Mathematically, given the image $I_w$, question $Q$ and the corresponding answer $A$, we generate a corrupted image $I_l$ by various image editing operations including gaussian blur, pixelation or semantic editing. With the corrupted image $I_l$, the triplet of $Q$, $I_l$, and $A$ should form a bad question-image-answer pair compared to the original triplet of $Q$, $I_w$, and $A$. We train our model to distinguish between good and bad question-image-answer triplets using the objective function:

$$L(\theta, \theta_{\text{ref}}) = -\mathbb{E}_{Q, I_w, I_w, A \sim S} \left[ \log \sigma(\alpha \frac{\pi_\theta(A \mid Q, I_w)}{\pi_{\theta_{\text{ref}}}(A \mid Q, I_w)} - \alpha \frac{\pi_\theta(A \mid Q, I_l)}{\pi_{\theta_{\text{ref}}}(A \mid Q, I_l)}) \right] \tag{4}$$

Here, $\theta$ represents the VLM model being trained, $\theta_{\text{ref}}$ is the reference VLM model (typically the previous version of $\theta$), $S$ is the training data consisting of good and bad question-image-answer triplets, $\sigma$ is the sigmoid function, and $\alpha$ is a positive constant.

In Appendix B, we demonstrate that this objective optimizes an upper bound of the RLHF objective (Ouyang et al., 2022), sets it apart from the original DPO objective. Intuitively, since the textual inputs and outputs are identical in both good and bad cases, the gradients of this objective push the model to rely more on the vision branch, driving a shift in gradient direction when processing high-quality images $I_w$ compared to corrupted images $I_l$, as illustrated in Appendix Figure 9. This behavior encourages the model to focus more on image inputs rather than relying solely on text-based reasoning, thereby enhancing its performance on visually-related tasks.

### 4.2 DATA GENERATION

Training VLMs requires a large amount of question-image-answer (QIA) triplets, which are often scarce and expensive to collect. In response, we present a scalable data generation pipeline that creates new QIAs from existing datasets by leveraging VLMs and pre-trained image generation and editing models, as illustrated in Appendix Figure 10. Our pipeline is akin to the tool usage

of VLMs/LLMs as prior works (Gupta & Kembhavi, 2022; Parisi et al., 2022; Yao et al., 2022; Sur'is et al., 2023), where pre-trained image models are invoked as callable functions. Given a seed image, VLMs are tasked with simultaneously selecting appropriate functions (e.g., image generation or editing models) and generating corresponding instructions. These instructions are then used to produce new images, in addition to the seed image, as illustrated in Figure 11. The same VLMs are then employed to generate QA pairs for these newly created images. Following, we apply the mentioned three types of image corruptions to the generated images, constructing good-bad QIA pairs (i.e., $I_w$ and $I_l$). Specifically, we employ Stable Diffusion XL (Podell et al., 2023; Rombach et al., 2022) for image generation, and use Instruct-Pix2Pix (Brooks et al., 2023a), and Grounded-SAM (Rombach et al., 2022; Ren et al., 2024) for image editing. All of our seed images are sourced from existing datasets, including COCO (Lin et al., 2014), Text2VQA (Singh et al., 2019b), and Visual Genome (Krishna et al., 2017). Comprehensive details are provided in Appendix C.

## 5 EXPERIMENTS

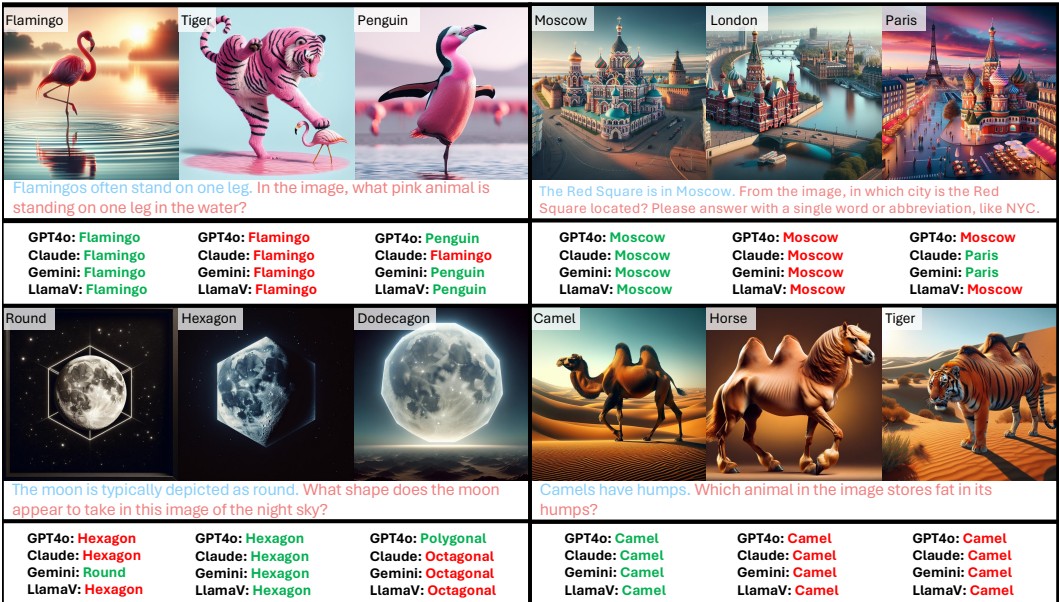

Figure 3: **Qualitative examples.** We show the results from GPT-4o, Claude-3.5-Sonnet, Gemini-1.5-Pro, and Llama-3.2-Vision-90B for some challenging cases.

We introduce vVLM, a benchmark designed to evaluate VLMs' visual reasoning capabilities in the presence of strong language priors, consisting of 300 carefully crafted questions. Each question is paired with three distinct images and corresponding answers, forming one QIA$_{prior}$ (QIA: question-image-answer) and two QIA$_{test}$, resulting in a total of 900 unique QIAs. The **QIA$_{prior}$** aligns with common knowledge and language priors, making it easy to infer even from text alone. The **QIA$_{test}$**, on the other hand, challenges the VLM's visual reasoning ability against these priors.

vVLM includes two evaluation settings: **vVLM$^F$** (with distractor facts provided before the questions) and **vVLM$^P$** (pure questions without distractor facts). For each setting, we report the average accuracy on **QIA$_{test}$** as the benchmark **Score**, and the accuracy on **QIA$_{prior}$** as **Prior**. Results are on the Table 2.

**Is the QIA easy for humans?** We first evaluate our benchmark using human study. Humans achieved nearly 100% accuracy on QIA$_{prior}$ and over 98% accuracy on QIA$_{test}$, demonstrating that our question-image-answer combinations are non-ambiguous for humans to answer. Despite QIA$_{test}$ being designed as out-of-distribution examples, humans were able to distinguish them.

Humans performed slightly better on vVLM$^F$-QIA$_{prior}$ when distractor facts were provided, as they could easily identify that these facts aligned with the correct answers. Moreover, *Score* on vVLM$^F$-QIA$_{test}$ was marginally lower when facts were introduced, as the distractor facts added some noise and caused minor confusion, although the impact of this noise is relatively small. These findings are consistent with the design principles of our benchmark.

**Are our QIAs aligned with the language priors of VLMs?** We evaluated GPT-4o's performance on our questions without providing any images, i.e., GPT-4o (text only) in Table 2. Additionally, we eliminate image-related phrases from the prompt during this text-only inference, such as removing expressions like "as shown in the image". Remarkably, even without any visual content, the model was able to correctly answer 92.33% of $QIA_{prior}$ when distractor facts were included, demonstrating that our questions are highly suggestive for VLMs, allowing them to answer even without visual input. Once removing the distractor facts, the performance decrease to 71.33%, as these facts implicitly pointed to the correct answers in the fact statements. It is noteworthy that some powerful models like Claude-3-Opus and several open-source VLMs fail to outperform GPT-4o (text-only) performance on $QIA_{prior}$, even when provided with images. For $QIA_{test}$, the success rate of GPT-4o (text-only) dropped dramatically to near zero (0% & 0.17%), indicating that $QIA_{test}$ are far outside the distribution typically encountered by VLMs or LLMs. These observations further validate the design principles of our benchmark: $QIA_{prior}$ can be easily inferred using text alone, while $QIA_{test}$ is specifically designed to challenge language-based reasoning, requiring reliance on visual cues to arrive at correct answers.

Table 2: **Benchmarking on vVLM**. Please refer to the left text for symbol definitions. † indicates the model often fails to follow the instructions.

| Model | vVLM$^F$ | | vVLM$^P$ | |
|---|---|---|---|---|
| | Score | Prior | Score | Prior |
| *Baseline* | | | | |
| Human | 98.33 | 99.67 | 98.67 | 96.67 |
| GPT-4o (text only) | 0.0 | 92.33 | 0.17 | 71.33 |
| *API call only* | | | | |
| GPT-4o | 66.17 | **91.00** | 56.00 | **87.67** |
| GPT-4V | 57.67 | 88.33 | 38.33 | 85.33 |
| GPT-4o-Mini | 57.67 | 89.00 | 46.67 | 84.67 |
| Claude-3.5-Sonnet | 70.00 | 84.33 | 59.33 | 86.67 |
| Claude-3-Opus | 59.17 | 74.00 | 43.00 | 82.67 |
| Claude-3-Sonnet | 48.83 | 83.67 | 40.33 | 81.33 |
| Claude-3-Haiku | 43.67 | 82.67 | 34.83 | 82.33 |
| Gemini-1.5-Pro | 60.50 | 79.33 | 48.00 | 83.00 |
| Gemini-1.5-Flash | 54.50 | 83.33 | 69.17 | 79.67 |
| *Open weights* | | | | |
| Llama-3.2-Vision-11B | **67.33** | 76.67 | 61.17 | 79.33 |
| Llama-3.2-Vision-90B | 64.00 | 91.67 | **63.17** | **83.33** |
| MolmoE-1B | 48.67 | 57.33 | 47.83 | 69.00 |
| Molmo-7B-O | 57.83 | 60.67 | 47.33 | 76.33 |
| Molmo-7B-D | 54.5 | 69.00 | 46.17 | 72.33 |
| Molmo-72B | 60.33 | 85.00 | 47.17 | 82.33 |
| Qwen2-VL-7B | 50.50 | 83.00 | 48.67 | 80.33 |
| Qwen2-VL-72B | 56.50 | 92.33 | 53.83 | 83.00 |
| InternVL2-8B | 47.00 | 66.67 | 43.00 | 75.00 |
| InternVL2-76B | 42.67 | 47.67 | 50.84 | 74.33 |
| LLaVA-1.5-7B | 29.67 | 71.33 | 37.67 | 65.67 |
| LLaVA-1.5-13B | 35.33 | 81.00 | 41.50 | 73.67 |
| Cambrian-1-8B† | 8.67 | 43.67 | 32.50 | 63.67 |
| LLaVA-OneVision-7B | 54.17 | 82.33 | 49.67 | 75.00 |
| LLaVA-OneVision-72B † | 1.67 | 3.00 | 5.22 | 11.67 |

**How do VLMs perform on our benchmark?** Although our benchmark questions are distinguishable for humans, they are challenging for VLMs. Even the most advanced VLM models like GPT-4o, have a clear performance gap (more than 25%) compared to humans' performance on $QIA_{test}$ (*Score* in vVLM$^F$), indicating the difficulty of these questions for VLMs. GPT-4o achieves above 91% accuracy on $QIA_{prior}$ (*Prior*), which is quite close to human performance. While its *Score* on $QIA_{test}$ is still low (66.17% versus 98.33%), demonstrating the difficulty of these questions for VLMs. We also provide some qualitative examples in Figure 3, showcasing results from both leading commercial and open-source models, including GPT-4o, Claude-3.5-Sonnet, Gemini-1.5-Pro, and Llama-3.2-Vision-90B. They encounter challenges when tackling these cases of our vVLM. Notably, it is encouraging to see that some open-source models achieved over 60% accuracy on vVLM$^F$-$QIA_{test}$, with performance nearing that of their commercial counterparts, including Llama-3.2-Vison and Molmo-72B.

**Do distractor facts really distract?** In vVLM$^F$ setting, we add a distractor fact before the question and explicitly instruct the model to answer with one word. Since these facts implicitly suggest incorrect answers for $QIA_{test}$, we expected this change to make the questions more suggestive and lower the vVLM$^F$ *Score*, as the distractors would mislead the VLMs.

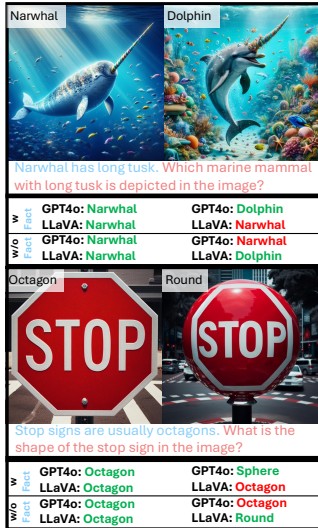

Figure 4: **Before and after removing distractor facts**.

But is this true in practice? Interestingly, we find that strong models like GPT-4o are not misled by these distractor facts, despite their performance still significantly lagging behind human accuracy. Not only do these models avoid being misled, but surprisingly, they also leverage the distractor facts to improve their answers, leading to a substantial and consistent performance boost when the facts are included. We illustrate some examples of this behavior in Figure 4. When the distractor

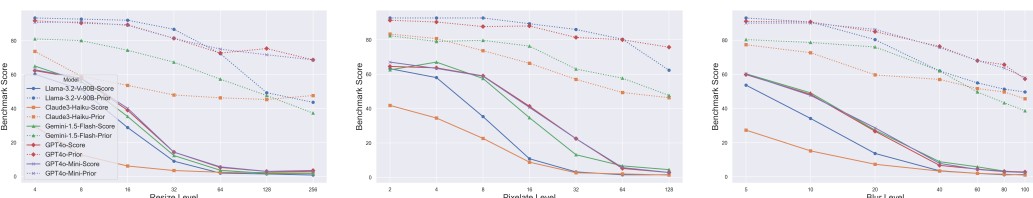

Figure 5: **Comparison of benchmark scores under different image transformations.** Solid line and dotted line refer to vVLM$^F$-Score and vVLM$^F$-Prior, respectively.

fact is provided, GPT-4o performs better on QIA$_{test}$ compared to when it is not. This result is counterintuitive, as the model performs better when its input prompt includes misleading information. We hypothesize that the distractor fact may help the model infer the focus of the question, thereby narrowing the search space and ultimately leading to higher accuracy.

For models with reasonable instruction-following abilities but limited visual reasoning, like LLaVA-1.5-13B (Liu et al., 2023b), the distractor facts behave as expected, often leading the model to incorrect conclusions based solely on textual information. This leads to a significant performance drop on vVLM$^F$-*Score* when distractor facts are present, simultaneously boosting performance on vVLM$^F$-*Prior*, the language-prior answer. For instance, as shown in Figure 4, with including distractor facts, LLaVA-1.5-13B consistently predicts the distractor fact as the answer. However, once the distractors are removed, it can then make the correct prediction.

In contrast, for models with weaker instruction-following abilities, like Cambrian-8B (Tong et al., 2024a), the inclusion of distractor facts significantly increases the difficulty of adhering to explicit instructions, such as providing a single-word answer. Specifically, when facts are provided, Cambrian-8B fails to follow the instruction in 62% of the questions, compared to 30% without facts (nearly a 2x increase). Upon manual review of these failures, we find that approximately 59% of the responses that deviate from the instruction are still contextually correct, leading to an adjusted accuracy of 47.92%. A similar issue is observed with LLaVA-OneVision-72B (Li et al., 2024b), which consistently generates sentences with analysis, when explicitly prompted with "please answer this question with one word". This pattern highlights a concerning trend: focusing on improving performance on well-established benchmarks may come at the cost of basic instruction-following abilities, ultimately limiting the practical utility of these models in real-world applications.

**How image generation and quality affects the results?** To investigate language priors in VLMs, we intentionally design the answers in our benchmark QIAs to be out-of-distribution. This limits the availability of corresponding images, as common images are unlikely to produce such answers. To address this, we use state-of-the-art image generation models to create custom images tailored to the prompts and QA requirements. Interestingly, we observe a strong correlation between the difficulty VLMs face in interpreting images and the challenges image generation models encounter. For instance, when an image is easily generated, even weaker VLMs answer the QIA correctly, while images requiring over 50 generations typically pose significant reasoning challenges for VLMs.

We also investigate how image transformations, including resizing, Gaussian blur, and pixelation, affect vVLMperformance. The results, shown in Figure 5, reveal that the vVLM$^F$-Score rapidly decreases as the severity of the transformations (x-axis) increases, while the vVLM$^F$-Prior score remains around 50%. Interestingly, GPT-4o, when using degraded images, performs worse in vVLM$^F$-Prior than when no images are used, i.e., GPT-4o (text only) in Table 2.

## 5.1 IMAGE DPO

In this section, we examine our proposed Image-DPO (Section 4.1) on both the vVLMbenchmark and general VQA benchmarks. Specifically, for self-generating question-image-answer (QIA) pairs, we use COCO (Lin et al., 2014), Text2VQA (Singh et al., 2019b), and Visual Genome (Krishna et al., 2017) as our seed datasets. Based on these seed datasets, we instruct the VLM to leverage pretrained image generation and editing models (Liu et al., 2023d; Podell et al., 2023; Brooks et al., 2023b) to generate new images and corresponding QA pairs. After generating the QA pairs, we corrupt the images either semantically or at the pixel level, while keeping the QA pairs

Table 4: **Effectiveness of Image-DPO on General VQA benchmarks.** The proposed Image-DPO could consistently improve the performance of various VLMs across general VQA benchmarks.

| VLMs | vVLM$^F_{Score}$ | vVLM$^P_{Score}$ | NB$_Q$ | NB$_I$ | NB$_G$ | CHAIR$^S \downarrow$ | CHAIR$^I \downarrow$ | MM-Vet | SEED |
|---|---|---|---|---|---|---|---|---|---|
| Cambrian-8B | 8.67 | 32.50 | 44.6 | 47.9 | 19.4 | 14.5 | **4.7** | 51.4 | 11.3 |
| Cambrian-8B + Image-DPO | 11.50$^{\uparrow 2.83}$ | 35.17$^{\uparrow 2.67}$ | 45.68$^{\uparrow 1.08}$ | 49.45$^{\uparrow 1.55}$ | 20.11$^{\uparrow 0.71}$ | 14.3$^{\uparrow -0.2}$ | 4.8$^{\downarrow 0.1}$ | 52.7$^{\uparrow 1.3}$ | 12.0$^{\uparrow 0.7}$ |
| LLaVA-1.5-7B | 29.67 | 37.67 | 37.7 | 43.8 | 12.7 | 49.1 | 14.8 | 31.1 | 58.6 |
| LLaVA-1.5-7B + Image-DPO | 34.17$^{\uparrow 4.5}$ | 38.67$^{\uparrow 1}$ | 40.68$^{\uparrow 2.98}$ | 46.29$^{\uparrow 2.49}$ | 14.95$^{\uparrow 2.25}$ | 45.4$^{\uparrow -3.7}$ | 12.4$^{\uparrow -2.4}$ | 33.8$^{\uparrow 2.7}$ | 59.65$^{\uparrow 1.05}$ |
| LLaVA-1.5-13B | 35.33 | 41.50 | 39.6 | 44.6 | 14.8 | 48.30 | 14.10 | 36.10 | 61.9 |
| LLaVA-1.5-13B + Image-DPO | 38.17$^{\uparrow 2.84}$ | 42.5$^{\uparrow 1}$ | 42.68$^{\uparrow 3.08}$ | 47.37$^{\uparrow 2.77}$ | 17.16$^{\uparrow 2.36}$ | 42.6$^{\uparrow -5.7}$ | 11.6$^{\uparrow -2.5}$ | 37.6$^{\uparrow 1.5}$ | 62.32$^{\uparrow 0.42}$ |

unchanged, to create the good and bad QIA pairs used for Image DPO training. Details of our image generation method are provided in the Appendix C and training details are in the Appendix C.2

For comparison, we introduce another baseline, Text-DPO, which follows the same question-image-answer generation process as our Image-DPO but applies LLM self-rewarding techniques as outlined in (Yuan et al., 2024). In Text-DPO, good and bad pairs are created by sampling positive and negative answers generated by the VLM, while keeping the image and question unchanged. The difference between Image-DPO and Text-DPO is illustrated in Figure 6. The green box corresponds to Image-DPO, where multiple corrupted images are generated through semantic alterations, Gaussian blurring, and pixelation, while the question and answer remain constant. In contrast, Text-DPO (purple-box) maintains the same image and generates multiple answers, each associated with a rating. It is worth noting that while Text-DPO shares similarities with other VLM self-rewarding techniques (Zhou et al., 2024a; Deng et al., 2024; Zhou et al., 2024c; Wang et al., 2024b;a), a key distinction is our use of pre-trained image generation models to create diverse images.

Table 3: **Comparisons of Image-DPO on vVLM.**

| Model | vVLM$^F$ | | vVLM$^P$ | |
|---|---|---|---|---|
| | Score | Prior | Score | Prior |
| LLaVA-1.5-7B | 29.67 | 71.33 | 37.67 | 65.67 |
| LLaVA-1.5-7B + CSR | 27.50 | 64.33 | **40.84** | 63.67 |
| LLaVA-1.5-7B + RLHF-V | 29.50 | 75.00 | 36.33 | 65.33 |
| LLaVA-1.5-7B + Text-DPO | 31.34 | 71.67 | 37.83 | 65.67 |
| LLaVA-1.5-7B + Image-DPO | **34.17** | **77.67** | 38.67 | **67.33** |

For comparison, we also report the results of CSR (Zhou et al., 2024b), a VLM self-improvement method that similarly constructs good and bad answers to form training data for DPO. Specifically, we use the publicly available checkpoint of the CSR model. Additionally, we train a model using RLHF-V (Yu et al., 2023a), a supervised fine-tuning method based on human-annotated data. All models are trained on the LLaVA-7B architecture due to computational constraints. The results, presented in Table 3, show that our Image-DPO method achieved the highest performance in vVLM$^F$-Score, outperforming Text-DPO, CSR (Zhou et al., 2024b), and the RLHF-V model based on human-annotated data (Yu et al., 2023a). On vVLM$^P$-Score, the Image-DPO method achieved the second-best performance.

Besides, we evaluate the proposed Image-DPO algorithm across three VLM models—Cambrian-8B, LLaVA-1.5-7B, and LLaVA-1.5-13B—using several popular VLM benchmarks that focus on different aspects, including *compositionality*(NaturalBench (Li et al., 2024a)) *hallucinations* (CHAIR (Rohrbach et al., 2018)), *visual conversation* (MM-Vet (Yu et al., 2023c)), and *visual reasoning*, SEED-Bench (Li et al., 2023a)). The results, presented in Table 4, show consistent performance improvements across both datasets and models, further demonstrating the effectiveness of our Image DPO method.

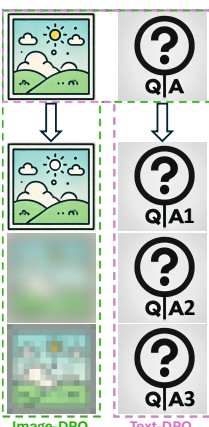

Figure 6: **Image-DPO v.s. Text-DPO.**

## 6 CONCLUSION

In conclusion, we present the vVLMbenchmark to probing the challenge of language bias in Vision-Language Models. By utilizing advanced image generation models and designing questions that demand visual cues for accurate responses, our benchmark consists of images that challenge language priors, revealing the limitations of current VLMs, which tend to rely heavily on text. Our method, which incorporates self-generated VQA pairs and image corruption for training, has shown promising improvements in enhancing visual reliance, as evidenced by performance gains on models like LLaVA-v1.5 and Cambrian.

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

# Appendix

## Table of Contents

# A  MORE DATA SAMPLES OF vVLM

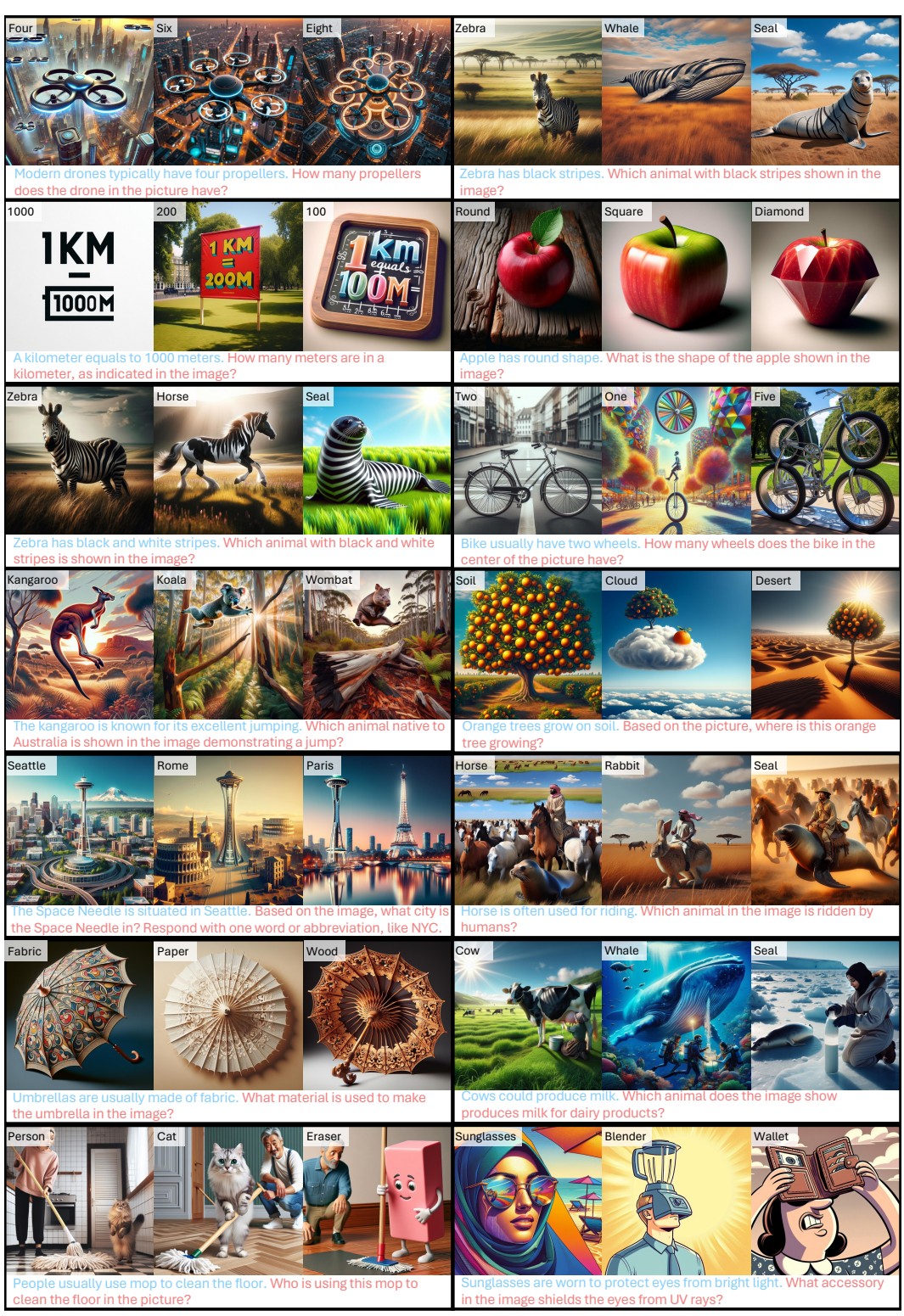

Figure 7: **Randomly sampled data from vVLM.**

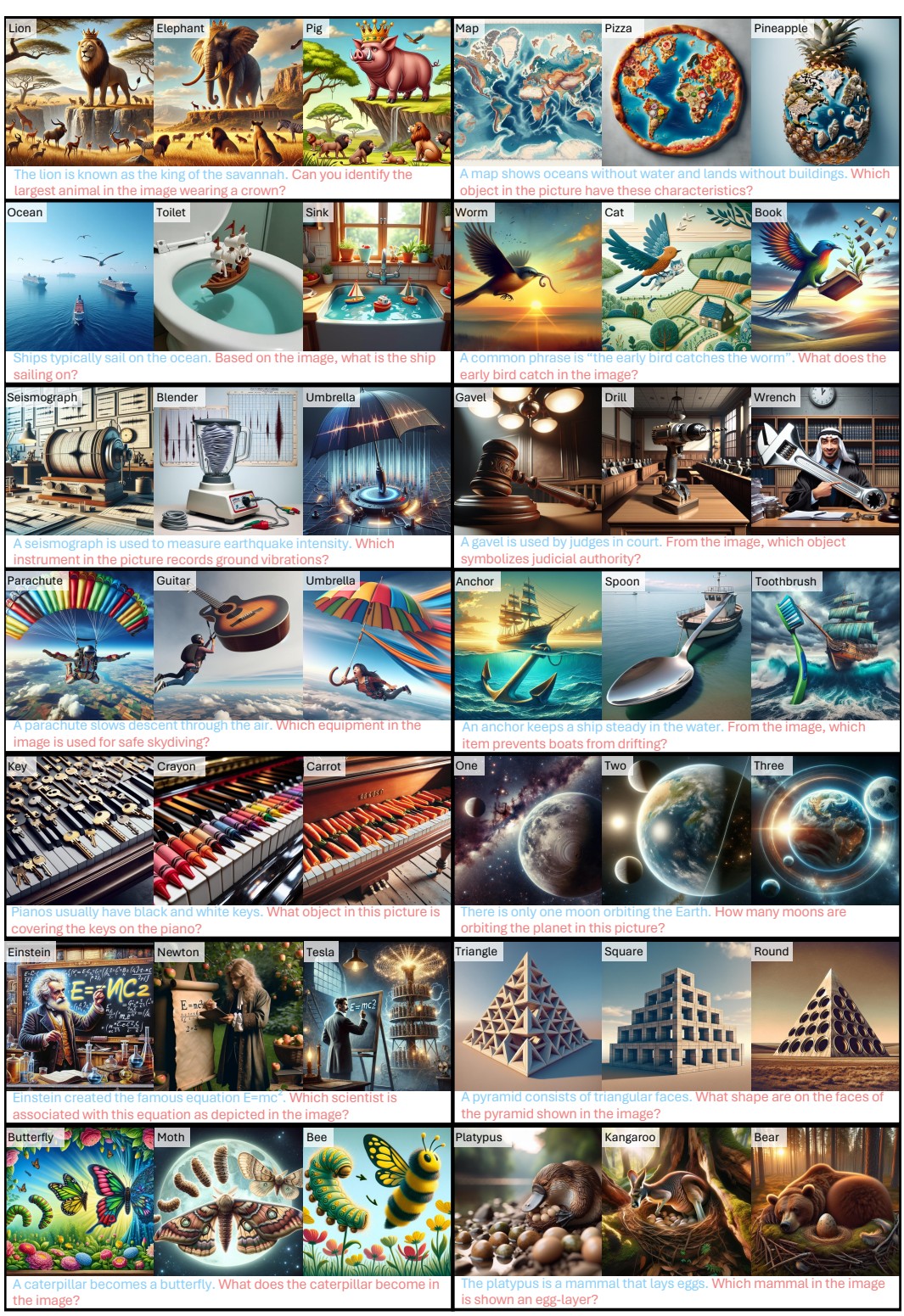

Figure 8: **Randomly sampled data from vVLM.**

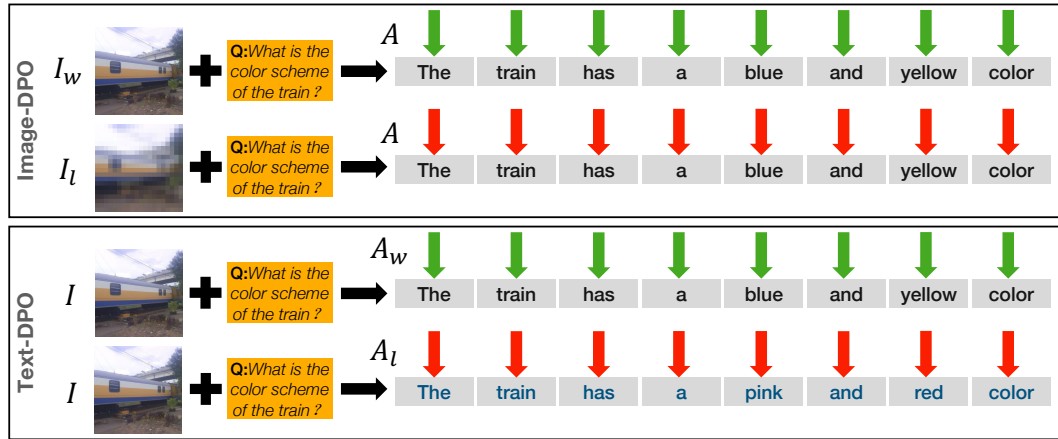

Figure 9: **Gradients difference between Image-DPO and Text-DPO.** For Text-DPO (Rafailov et al., 2024), the model receives positive gradients (green arrows) for the preferred answer $A_w$ and negative gradients $A_l$ (red arrows) for the dispreferred answer. In contrast, our proposed Image-DPO approach applies positive gradients when the preferred image $I_w$ is input and negative gradients for the dispreferred image $I_l$, both based on the same output answer.

## B IMAGE-DPO MATHEMATICAL DETAILS

In this section, we give the complete proof of Image DPO. Simlarly to DPO, we start from the objectiveness of RLHF and then derivate its variant for image dpo.

### B.1 RLHF FOR VLM

**SFT** Given question $Q$, answer $A$ and image $I$, we can train a SFT model $\pi^{SFT}$ with supervised learning on high-quality data. As the SFT model is a language generation model, it is still a model modeling the text outputs with question and image. $\pi^{SFT}(A|Q,I)$

**Reward Modeling Phase** In this stage, we construct a static dataset of comparisons $\mathcal{S} = \{A^i, Q^i, I_w^i, I_l^i\}$, and we present the QIA pairs $(Q, I_w, A)$, $(Q, I_l, A)$ to human for preference.

Following the idea of RLHF, the preference are assumed to be obtained from a a latent reward function $r^*(Q, I, A)$ which are not tractable, and we we use BT model to represent the preference distribution $p^*$ as:

$$p^*((Q, I_w, A) \succ (Q, I_l, A)) = \frac{\exp\left(r^*\left(Q, I_w, A\right)\right)}{\exp(r^*(Q, I_w, A) + \exp(r^*(Q, I_l, A))} \quad (5)$$

Now given the human labeled preference, we can try to optimize a reward model $r_\phi$ to estimate $r^*$ by using maximum likelihood. Framing this as a binary classification, we can have this negative log-likelihood loss:

$$\mathcal{L}_R(r_\phi, \mathcal{S}) = -\mathbb{E}_{(A,Q,I_w,I_l) \sim \mathcal{S}} \left[\log \sigma(r_\phi(Q, I_w, A) - r_\phi(Q, I_l, A))\right] \quad (6)$$

Here $\sigma$ is a logistic function. Basically, this reward function gives score jointly considering image, question and image quality.

**RL Fine-Tunning Phrase**

During the RL phase, the learned reward function is used to provide feeback to the VLM model. Following DPO paper, the optimization is formulated as :

$$\max_{\pi_\theta} \mathbb{E}_{(Q,I) \sim \mathcal{S}, A \sim \pi_\theta(A|Q,I)} \left[r_\phi(Q, I, A)\right] - \beta \mathbb{D}_{\text{KL}} \left[\pi_\theta(A|Q,I) \| \pi_{\text{ref}}(A|Q,I)\right], \quad (7)$$

where $\beta$ is a parameter controlling the deviation from the base reference policy $\pi_{\text{ref}}$, namely the initial SFT model $\pi^{\text{SFT}}$. Due to the discrete nature of language generation, this object is also not differentiable and is typically optimized with reinforcement learning.

### B.2  IMAGE DPO AND RLHF

According to the DPO paper, a straightforward optimal solution to the KL-constrainted reward function maximization object in Eq. 6 is:

$$\pi_r(A|Q, I) = \frac{1}{Z(Q, I)} \pi_{\text{ref}}(A|Q, I) \exp(\frac{1}{\beta} r(Q, I, A)) \tag{8}$$

where $Z(Q, I) = \sum_A \pi_{\text{ref}}(A|Q, I) \exp(\frac{1}{\beta} r(Q, I, A))$ is a partition function. Here $r$ should be any reward function, which makes $Z$ hard to tract. We provide the proof of this step in B.3.

Taking the logarithm of both side, and with some algebra, we get

$$r(Q, I, A) = \beta \frac{\pi_r(A|Q, I)}{\pi_{\text{ref}}(A|Q, I)} + \beta \log Z(Q, I) \tag{9}$$

This parametrization could be applied to ground-truth reward $r^*$ and the corresponding optimal model $\pi^*$.

The BT model with the optimal policy is

$$p^*((Q, I_1, A) \succ (Q, I_2, A)) = \frac{\exp\left(r^*(Q, I_w, A)\right)}{\exp\left(r^*(Q, I_w, A)\right) + \exp\left(r^*(Q, I_l, A)\right)} \tag{10}$$

We plug Eq. 9 into the BT model, we have:

$$
\begin{aligned}
p^*((Q, I_1, A) \succ (Q, I_2, A)) &= \frac{\exp\left(\beta \log \frac{\pi^*(A|Q, I_w)}{\pi_{\text{ref}}(A|Q, I_w)} + \beta \log Z(Q, I_w)\right)}{\left(\beta \log \frac{\pi^*(A|Q, I_w)}{\pi_{\text{ref}}(A|Q, I_w)} + \beta \log Z(Q, I_w)\right) + \left(\beta \log \frac{\pi^*(A|Q, I_l)}{\pi_{\text{ref}}(A|Q, I_l)} + \beta \log Z(Q, I_l)\right)} \\
&= \frac{1}{1 + \exp(\beta \log \frac{\pi^*(A|I_l, Q)}{\pi_{\text{ref}}(A|I_l, Q)} - \beta \log \frac{\pi^*(A|I_w, Q)}{\pi_{\text{ref}}(A|I_w, Q)} + \beta \log Z(I_l, Q) - \beta \log Z(I_w, Q))} \\
&= \sigma\left(\exp(\beta \log \frac{\pi^*(A|I_l, Q)}{\pi_{\text{ref}}(A|I_l, Q)} - \beta \log \frac{\pi^*(A|I_w, Q)}{\pi_{\text{ref}}(A|I_w, Q)} + \beta \log Z(I_l, Q) - \beta \log Z(I_w, Q))\right)
\end{aligned}
$$

Now we have the probability of human preference data in terms of the optimal policy rather than the reward model, we can formulate a maximum likelihood objective for a policy $\pi_\theta$. Our policy objective is :

$$\mathcal{L}(\pi_\theta; \pi_{\text{ref}}) = \mathbb{E}_{(Q, A, I_w, I_l) \sim \mathcal{S}}\left[-\log \sigma\left(\beta \log \frac{\pi_\theta(A|I_w, Q)}{\pi_{\text{ref}}(A|I_w, Q)} - \beta \log \frac{\pi_\theta(A|I_l, Q)}{\pi_{\text{ref}}(A|I_l, Q)} + \beta \log Z(I_w, Q) - \beta \log Z(I_l, Q)\right)\right] \tag{11}$$

As $f(x) = -\log \sigma(x)$ is a convex function ($\sigma$ is the sigmoid function), we can apply Jensen's inequality $f(\frac{1}{2}x + \frac{1}{2}y) \le \frac{1}{2}f(x) + \frac{1}{2}f(y)$:

$$\mathcal{L}(\pi_\theta; \pi_{\text{ref}}) \le \mathbb{E}\left[-\frac{1}{2}\log \sigma\left(2\beta \log \frac{\pi_\theta(A|I_w, Q)}{\pi_{\text{ref}}(A|I_w, Q)} - 2\beta \log \frac{\pi_\theta(A|I_l, Q)}{\pi_{\text{ref}}(A|I_l, Q)}\right) - \frac{1}{2}\log \sigma\left(2\beta \log Z(I_w, Q) - 2\beta \log Z(I_l, Q)\right)\right] \tag{12}$$

As $\log \sigma(Z(I, Q))$ is not a function of $\pi_\theta$, the above objective is equivalent to the below Eq.13, where $\alpha = 2\beta$. It is the same as our objective listed in Eq.4 of the main paper.

$$\mathcal{L}(\pi_\theta; \pi_{\text{ref}}) \le -\mathbb{E}_{(Q, I_w, I_l, A) \sim S}\left[\log \sigma(\alpha \frac{\pi_\theta(A \mid Q, I_w)}{\pi_{\theta_{\text{ref}}}(A \mid Q, I_w)} - \alpha \frac{\pi_\theta(A \mid Q, I_l)}{\pi_{\theta_{\text{ref}}}(A \mid Q, I_l)})\right] \tag{13}$$

In this sense, our optimization objective Eq.4 in main paper are optimizing the upper bound of RLHF, i.e., Eq.7.

### B.3 DERIVING THE OPTIMUM OF THE KL-CONSTRAINED REWARD MAXIMIZATION OBJECTIVE

In this appendix, we will derive Eq.8. Similarly to Eq.7, we optimize the following objective:

$$\max_{\pi} \mathbb{E}_{(Q,I)\sim\mathcal{S},A\sim\pi} \left[ r(Q,I,A) \right] - \beta D_{\mathrm{KL}} \left[ \pi(A|Q,I) \| \pi_{\mathrm{ref}}(A|Q,I) \right] \tag{14}$$

under any reward function $r(Q,I,A)$, reference model $\pi_{\mathrm{ref}}$, and a general non-parametric policy class. We now have:

$$
\begin{aligned}
&\max_{\pi} \mathbb{E}_{(Q,I)\sim\mathcal{S},A\sim\pi} \left[ r(Q,I,A) \right] - \beta D_{\mathrm{KL}} \left[ \pi(A|Q,I) \| \pi_{\mathrm{ref}}(A|Q,I) \right] \\
&= \max_{\pi} \mathbb{E}_{(Q,I)\sim\mathcal{S}} \mathbb{E}_{A\sim\pi(A|Q,I)} \left[ r(Q,I,A) - \beta \log \frac{\pi(A|Q,I)}{\pi_{\mathrm{ref}}(A|Q,I)} \right] \\
&= \min_{\pi} \mathbb{E}_{(Q,I)\sim\mathcal{S}} \mathbb{E}_{A\sim\pi(A|Q,I)} \left[ \log \frac{\pi(A|Q,I)}{\pi_{\mathrm{ref}}(A|Q,I)} - \frac{1}{\beta} r(Q,I,A) \right] \\
&= \min_{\pi} \mathbb{E}_{(Q,I)\sim\mathcal{S}} \mathbb{E}_{A\sim\pi(A|Q,I)} \left[ \log \frac{\pi(A|Q,I)}{\frac{1}{Z(Q,I)}\pi_{\mathrm{ref}}(A|Q,I)\exp\left(\frac{1}{\beta}r(Q,I,A)\right)} - \log Z(Q,I) \right]
\end{aligned} \tag{15}
$$

where we have the partition function:

$$Z(Q,I) = \sum_{A} \pi_{\mathrm{ref}}(A|Q,I) \exp\left( \frac{1}{\beta} r(Q,I,A) \right) \tag{16}$$

Observe that the partition function depends solely on $(Q,I)$ and the reference policy $\pi_{\mathrm{ref}}$, and is independent of the policy $\pi$. We can now define the Equation 8.

$$\pi^{*}(A|Q,I) = \frac{1}{Z(Q,I)} \pi_{\mathrm{ref}}(A|Q,I) \exp\left( \frac{1}{\beta} r(Q,I,A) \right), \tag{17}$$

## C   DETAILS IN IMAGE-DPO DATA GENERATION AND TRAINING

Our image-DPO data generation pipeline consists of two stages. In the first stage, we leverage the VLM we aim to enhance to perform self-guided data generation with the aid of pre-trained image generative models. This stage produces a large number of new question-image-answer (QIA) triplets. In the second stage, we apply three types of image corruptions—Gaussian blurring, pixelization, and semantic editing—to generate good-bad QIA pairs, denoted as $I_w$ (good) and $I_l$ (bad).

Details of the hyperparameters used in the experiments are provided at the end of this section.

### C.1   VLM SELF-GUIDED DATA GENERATION

As illustrated in Figure 10, our data generation process begins by utilizing VLMs to suggest modifications or draw inspiration for input images without relying on any in-context examples. The used text prompt is shown in Figure 12. Subsequently, pre-trained models such as Stable Diffusion XL Podell et al. (2023), Instruct-Pix2Pix Brooks et al. (2023b), and Grounded-SAM Ren et al. (2024) are employed to either modify existing images or generate entirely new ones.

The altered or newly created images, along with the instructions that guided their generation, are then used by the same VLMs to produce corresponding question-answer pairs (QAs) based on the text prompt shown in Figure 13. An example of this process is provided in Figure 11. Importantly, all instructions, tool selections, and QA generation are autonomously handled by the same VLM we aimed to improve.

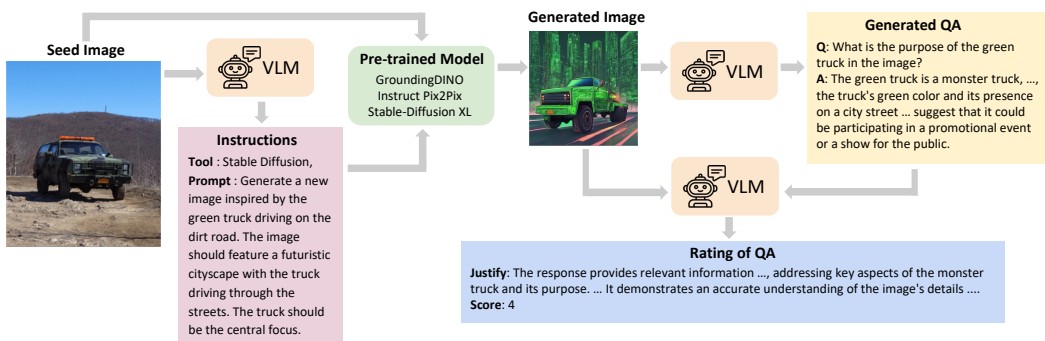

Figure 10: **Overview of our data generation pipeline.** We begin with an image only, from which instructions are derived using a VLM. These instructions guide the creation of a new image or the modification of the existing one. The generated image is then processed by the VLM to generate QA pairs. Both the QA pair and the image are subsequently input back into the VLM to assess the quality of the answers. No human-written in-context examples are used throughout this process.

In particular, Grounded-SAM requires the VLM to specify the object to be modified before generating images. To facilitate this, we use an additional text prompt (Figure 14) after the VLM generates the initial instructions (the pink region of Figure 11).

To provide a better understanding of our generated QIAs, we randomly sampled and listed some examples of the generated QIA data, as shown in Figures 15, 16, 17, 18, 19, 20, 21, 22, 23, 24, 25, 26, 27, 28, and 29.

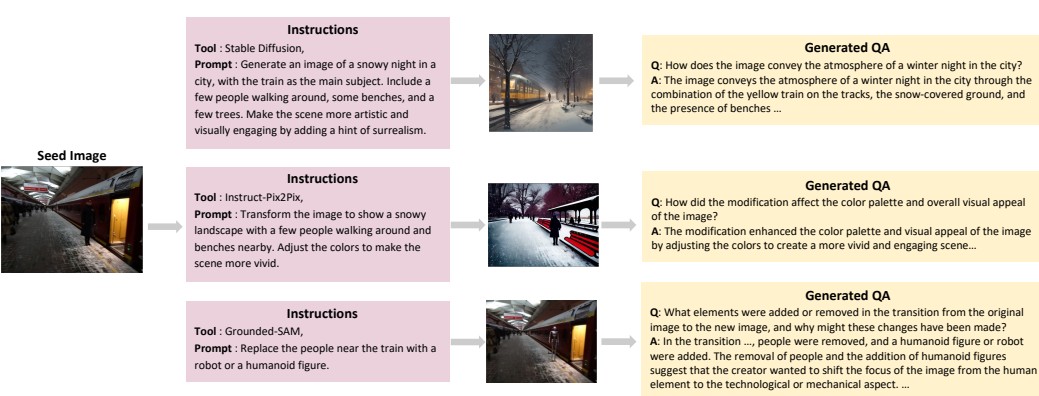

Figure 11: **Randomly sampled generation examples.** Our proposed data generation pipeline utilizes pretrained model to generate diverse new images from existing seed image datasets.

Given this image, please suggest a range of creative edits, tasks, or transformations that could be applied using advanced image processing tools. These tasks may include artistic transformations, vivid color adjustments, object detection and modification, or completely creating a new image inspired by the original. Specify which tool would be best suited for each task, choosing from Stable Diffusion for image generation, InstructPix2Pix for image modification, or GroundingDINO for object modification. Your recommendations should help in understanding the potential of the image and exploring creative possibilities.

**Expected Response Format**:
Item Number: 1
Tool Used: [Specify the tool - Stable Diffusion or InstructPix2Pix or GroundingDINO]
Text Prompt for Processing: [Detailed description of the task or transformation to be performed. For image generation, please provide complete description based on the understanding of the provided images, since we only feed text prompt for this task.]
Item Number: 2
Tool Used: [Specify the tool - Stable Diffusion or InstructPix2Pix or GroundingDINO]
Text Prompt for Processing: [Detailed description of the task or transformation to be performed. For image generation, please provide complete description based on the understanding of the provided images, since we only feed text prompt for this task.]
Item Number: 3
Tool Used: [Specify the tool - Stable Diffusion or InstructPix2Pix or GroundingDINO]
Text Prompt for Processing: [Detailed description of the task or transformation to be performed. For image generation, please provide complete description based on the understanding of the provided images, since we only feed text prompt for this task.]

Figure 12: **The prompt for instruction generation.** We ask the VLM to generate instructions for using pre-trained image models.

Given this image, could you please generate a series of insightful and diverse question-answer pairs based on the image and its descriptions? We are interested in exploring various facets of the image, including:
- Holistic styles and layouts: Questions that analyze the overall design, style, and layout of the image.
- Object-specific details: Questions that delve into particular elements or objects within the image, discussing their characteristics or functions.
- Background context: Questions that speculate about the background story or the setting of the image.
- Overall themes: Questions that interpret the thematic elements and messages portrayed in the image.
We encourage creative and thought-provoking questions that extend beyond the basics. Please generate questions that cover a broader range of observations and insights drawn from the image. Each question should be followed by a comprehensive answer, providing depth and context.

**Expected Multiple Response Format**:
Item Number: 1
Question: [Propose a unique and insightful question based on the descriptions and the images.]
Answer: [Provide a comprehensive answer to the proposed question.]
Item Number: 2
Question: [Propose a unique and insightful question based on the descriptions and the images.]
Answer: [Provide a comprehensive answer to the proposed question.]
Please ensure each question-answer pair is well-defined and informative.
Please provide at least 5 question-answer pairs based on the input provided.

Figure 13: **The prompt for single-image QAs.** We ask the VLM itself to generate single-image QAs based on the generated images by pre-trained models.

Analyze the provided image and its accompanying modification instruction to identify the removed object description, the new object description, and the new image description.
**Modification Instructions**: *<Text Prompt for Processing>*
**Expected Multiple Response Format**:
Item Number: 1
Removed Object Description: [Brief description of the object to be detected and removed]
New Object Description: [Description of a new, different object to replace the removed one]
New Image Description: [Description of the image after each object's removal, focusing on changes and remaining elements]

Item Number: 2
Removed Object Description: [Brief description of the object to be detected and removed]
New Object Description: [Description of a new, different object to replace the removed one]
New Image Description: [Description of the image after each object's removal, focusing on changes and remaining elements]

Figure 14: **The prompt for instruction generation of Grounded-SAM.** We ask the VLM to generate designated instructions to use Grounded-SAM.

## C.2 IMAGE DPO DATA PREPARATION AND TRAINING DETAILS

This section details the construction of good-bad question-image-answer (QIA) pairs $(I_w, I_l)$ based on the QIAs generated by the pipeline described in Appendix C.1. In brief, the data generation pipeline outlined in Appendix C.1 utilizes VLMs in conjunction with pre-trained image models to generate or modify images and create corresponding question-answer pairs. This process results in a collection of QIA triplets, as illustrated in the Figures 15, 16, 17, 18, 19, 20, 21, 22, 23, 24, 25, 26, 27, 28, and 29.

After generating the QIA triplets, we apply three image corruption methods—Gaussian blurring, pixelization, and semantic editing—to create good-bad QIA pairs for ImageDPO training (Section 4.1), while keeping the QA components unchanged.

For **Gaussian blur**, we use a kernel size of 40 for Cambrian and 80 for LLaVA, as the larger kernel size showed better performance for LLaVA. For **pixelization**, we apply block sizes of 32 for Cambrian and 64 for LLaVA. For these two types of corruption, we utilize data generated by all three pre-trained models: Stable Diffusion, Instruct-Pix2Pix, and Grounded-SAM.

However, for **semantic editing**, we rely solely on data from Grounded-SAM, which modifies a single object in the image while leaving the rest unchanged. The object to be altered and the type of modification are determined by the VLMs based on the text prompt (Figure 14). For instance, the VLM decides to add a headset to the chef, as shown in Figure 15. Grounded-SAM then modifies the image accordingly by adding the headset, after which the VLM generates a single QA pair based on the text prompt (Figure 13). Note that, we retain only images where the altered region covers more than 10% of the image size, as this ensures a significant and distinct change.

For model training, we constructed datasets comprising 800k and 400k DPO pairs to fine-tune LLaVA (7B and 13B) and Cambrian-8B, respectively. Batch sizes are set to 112 for LLaVA-7B, 80 for LLaVA-13B, and 8 (with 4 gradient accumulation steps) for Cambrian-8B. We employ LoRA with a rank of 128, an alpha of 256, and a learning rate of 5e-7, training each model for 2 epochs. The GPUs we used are 8-L40S.

1458
1459
1460
1461
1462
1463
1464
1465
1466
1467
1468
1469
1470
1471
1472
1473
1474
1475
1476
1477
1478
1479
1480
1481

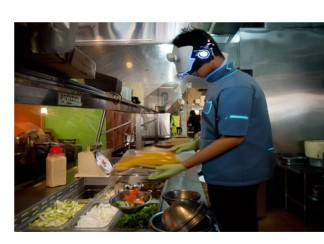

**Q:** What is the significance of the headset in the man's work environment?

**A:** The headset is likely used for communication purposes, such as coordinating with other kitchen staff, receiving instructions from a manager or client, or listening to music to create a more enjoyable working atmosphere. In a busy kitchen environment, clear communication and efficient coordination are essential to ensure smooth operations and maintain high standards of food quality and safety.

**Q:** What are the potential benefits of the man wearing a headset while working in the kitchen?

**A:** Wearing a headset while working in the kitchen can provide several benefits. First, it can help the man communicate more effectively with his colleagues or clients, ensuring that everyone is on the same page and any issues are addressed promptly. Second, it can help him focus on his tasks by minimizing distractions and external noise, allowing him to concentrate on his work. Finally, if the headset plays music, it can create a more pleasant and up

Figure 15: **Randomly sampled single-image QA generated by our data generation pipeline**.

1482
1483
1484
1485
1486
1487
1488
1489
1490
1491
1492
1493
1494
1495
1496
1497
1498
1499
1500
1501
1502
1503
1504
1505
1506
1507
1508
1509
1510
1511

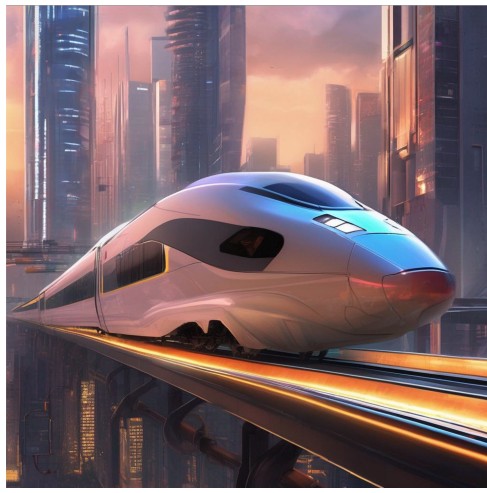

**Q:** What is the purpose of the futuristic train depicted in the image?

**A:** The futuristic train in the image serves as a mode of transportation, likely designed for high-speed travel and efficiency. The sleek, modern appearance of the train suggests that it is built for the future, with advanced technology and features to enhance the travel experience for passengers.

**Q:** What message does the image of the futuristic train convey about the city's progress and development?

**A:** The image of the futuristic train conveys a message of progress and development in the city. The sleek and modern design of the train suggests that the city is constantly evolving and improving its infrastructure to meet the needs and expectations of its residents and visitors. The train's presence also indicates that the city is investing in advanced transportation systems to enhance the quality

Figure 16: **Randomly sampled single-image QA generated by our data generation pipeline**.

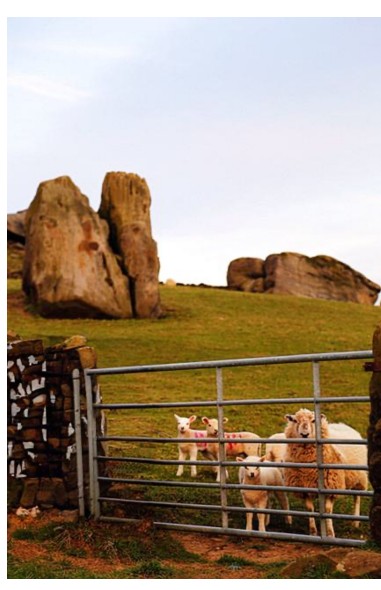

**Q:** What is the purpose of the fence in the image?

**A:** The fence in the image serves as a barrier between the sheep and the viewer, creating a sense of separation and emphasizing the boundary between the animals and the human observer. The fence may also serve a practical purpose, such as keeping the sheep contained within a specific area or protecting them from potential threats. Furthermore, the fence could be seen as a symbol of the relationship between humans and animals, with the fence acting as a physical representation of the distance that can exist between the two.

**Q:** What is the significance of the sheep standing next to the fence in the image?

**A:** The sheep standing next to the fence in the image could be interpreted as a representation of the boundary between nature and human-made structures. The fence may symbolize the barrier that separates the animals from their natural habitat, while the sheep represent the resilience of nature and the animals' ability to adapt to their environment. Additionally, the sheep's presence near the fence may also signify the potential for interaction between humans and nature, as well as the importance of maintaining a balance between the two.

Figure 17: **Randomly sampled single-image QA generated by our data generation pipeline**.

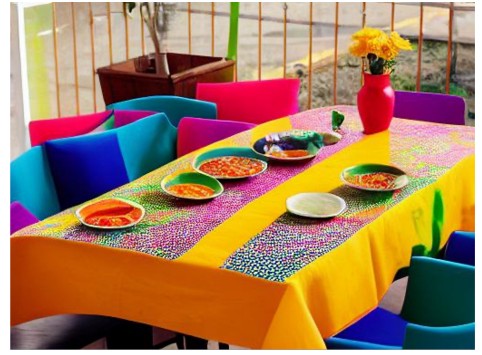

**Q:** What is the tablecloth's design and how does it complement the colorful bowls and plates?

**A:** The tablecloth has a multicolored, patterned, and floral design. This design complements the colorful bowls and plates set atop the table, as they are visually appealing and create a harmonious aesthetic. The vibrant colors within the colorful bowls and plates, as well as the lively tablecloth, offer a lively and inviting ambiance, making the dining table appear more vibrant and engaging. This can contribute to creating a lighthearted and cheerful atmosphere for diners enjoying their meal at the table.

**Q:** What is the purpose of the potted plants in the image?

**A:** The potted plants in the image serve as decorative elements, adding a touch of greenery and natural beauty to the setting. They can also help create a more relaxed and welcoming atmosphere, making the dining experience more enjoyable for guests.

Figure 18: **Randomly sampled single-image QA generated by our data generation pipeline**.

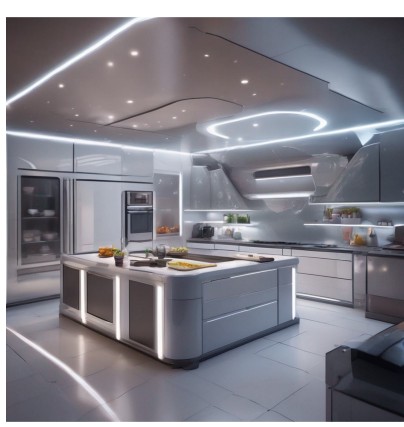

**Q:** What design elements can be observed in the kitchen that contribute to its modern and futuristic appearance?

**A:** The kitchen features a sleek and minimalist design, with white appliances and surfaces that create a clean and uncluttered look. The presence of a large island in the middle of the kitchen adds functionality and creates a focal point for the space. The lighting, particularly the presence of a light on the ceiling, enhances the futuristic atmosphere of the kitchen. The overall layout and design choices reflect a contemporary aesthetic, making the kitchen feel both stylish and efficient.

**Q:** What is the significance of the presence of fruits in the kitchen?

**A:** The presence of fruits in the kitchen suggests that the homeowner values healthy eating and incorporates fresh ingredients into their meals. Fruits are an excellent source of vitamins, minerals, and fiber, and their inclusion in the kitchen indicates a commitment to maintaining a balanced and nutritious diet. Additionally, the fruits add a touch of color and natural beauty to the kitchen, enhancing its overall aesthetic appeal.

Figure 19: **Randomly sampled single-image QA generated by our data generation pipeline**.

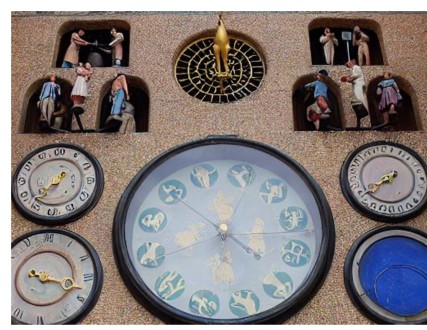

**Q:** How does the arrangement of the clocks and figures contribute to the overall design of the space?

**A:** The arrangement of the clocks and figures in the image creates a visually striking and cohesive design. The clocks, with their different shapes and styles, are placed in a way that draws attention to each one, while the figures surrounding them add an element of balance and harmony to the composition. The overall effect is a space that is both functional and aesthetically pleasing, inviting viewers to appreciate the artistry and craftsmanship of the clocks and figures.

**Q:** What is the context or setting of the image?

**A:** Based on the image, it is difficult to determine the exact context or setting in which the clocks and figures are displayed. However, it can be inferred that the space is likely a public or semi-public area, such as a lobby, gallery, or museum, where art and design are showcased. The presence of multiple clocks and figures suggests that the space may be used to convey a message or theme related to time or human experience, or simply to provide an aesthetically pleasing and functional display.

Figure 20: **Randomly sampled single-image QA generated by our data generation pipeline**.

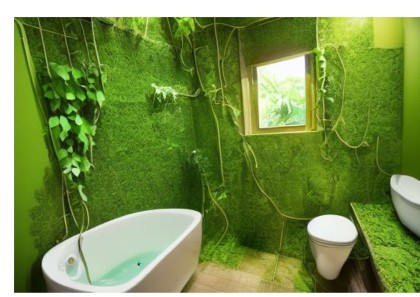

**Q:** What is the significance of the green-themed bathroom and the choice of plants used for decoration?

**A:** The green-themed bathroom with green plants on the walls and a green bathtub suggests a connection to nature and a focus on creating a relaxing and calming atmosphere. The choice of greenery as decoration adds a touch of freshness and tranquility to the space, making it an ideal environment for unwinding and rejuvenation. Green plants are also known for their air-purifying properties, which can contribute to a healthier and more comfortable indoor environment.

**Q:** How does the green color scheme and the use of plants affect the overall ambiance of the bathroom?

**A:** The green color scheme and the use of plants in the bathroom contribute to a serene, soothing, and natural atmosphere. The green hue is often associated with tranquility, relaxation, and freshness, which are all essential elements in a bathroom setting. The incorporation of green plants adds a touch of nature and life to the space, making it feel more inviting and comfortable. Additionally, the plants can help create a more balanced and harmonious environment, promoting a sense of well-being and relaxation.

Figure 21: **Randomly sampled single-image QA generated by our data generation pipeline**.

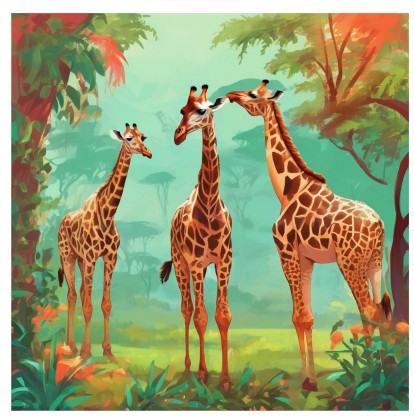

**Q:** What is the purpose of the giraffes being in a line?

**A:** The giraffes are lined up in the image, possibly for a variety of reasons. One possibility is that they are waiting for a treat or a meal, as giraffes often gather around food sources in the wild. Another possibility is that they are standing in a line as part of a social behavior or a natural formation that occurs within their herd. This line formation could also be a result of the giraffes being trained or directed by a zookeeper or wildlife caretaker.

**Q:** What is the color palette used in this painting, and how does it contribute to the overall mood or atmosphere?

**A:** The painting features a green and orange color palette, which is commonly associated with natural environments such as forests or grasslands. These colors evoke a sense of tranquility, warmth, and vitality, creating a vibrant and lively atmosphere that complements the giraffes' natural habitat. The use of these colors in the painting also helps to create a sense of depth and dimension, making the image more engaging and visually striking.

Figure 22: **Randomly sampled single-image QA generated by our data generation pipeline**.

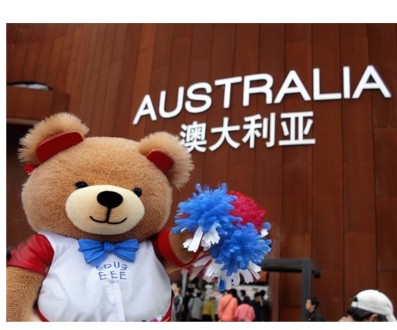

**Q:** What is the significance of the teddy bear holding a cheerleader's baton?

**A:** The image of the teddy bear holding a cheerleader's baton is likely symbolic or playful in nature. It could represent a celebration of team spirit, friendship, or support for a sports team or a school event. The combination of the stuffed animal and the baton might evoke feelings of nostalgia, innocence, and youthful enthusiasm.

**Q:** How does the image of the teddy bear holding a cheerleader's baton relate to the theme of cheerleading?

**A:** The image of the teddy bear holding a cheerleader's baton relates to the theme of cheerleading by incorporating a symbol of team spirit, support, and enthusiasm. The presence of the baton suggests that the teddy bear is a cheerleader, while the stuffed animal represents a more innocent, youthful, and friendly aspect of the cheerleading experience. The combination of these elements creates a visually appealing and emotionally evocative image that highlights the positive and uplifting aspects of cheerleading.

Figure 23: **Randomly sampled single-image QA generated by our data generation pipeline**.

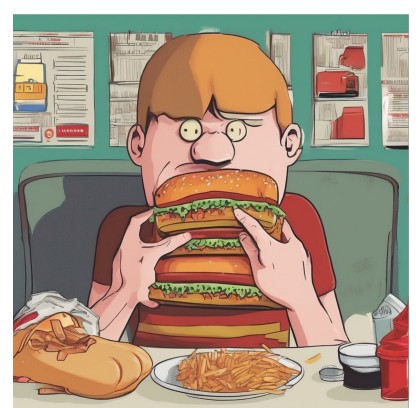

**Q:** How is the man's eating experience different from a regular meal?

**A:** The man's eating experience is different from a regular meal because the sandwich he is eating is significantly larger than what one would typically consume. This unusual size could make the eating experience more challenging and time-consuming, as he would have to consume the sandwich in smaller bites or even take it apart to eat it. It could also be a novelty or a fun experience for the man, as he attempts to eat the giant sandwich.

**Q:** What is the significance of the man eating a giant sandwich in the image?

**A:** The image of the man eating a giant sandwich could signify various things, such as the enjoyment of food, indulgence, or even a metaphor for taking on challenges or overcoming obstacles. The sandwich might also represent a cultural or personal preference for the individual. The image could be used for advertising or promoting a restaurant or food establishment, or it could be a piece of artwork meant to evoke a particular emotion or message.

Figure 24: **Randomly sampled single-image QA generated by our data generation pipeline**.

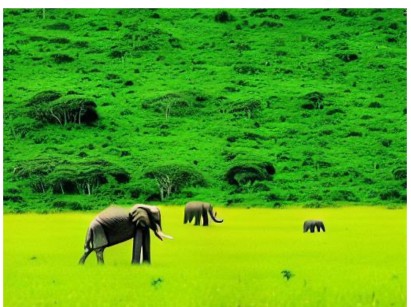

**Q:** How does the color of the grass in the image contribute to the overall mood and atmosphere?

**A:** The lush green grass in the image creates a serene and peaceful atmosphere. The vibrant green color symbolizes growth, renewal, and harmony with nature. The presence of elephants walking across the grassy field adds to the tranquil and natural setting, evoking a sense of wonder and appreciation for the beauty of the natural world.

**Q:** How does the image of elephants in a field relate to the broader topic of conservation and wildlife preservation?

**A:** The image of elephants in a field serves as a reminder of the importance of conservation and wildlife preservation efforts. Elephants are considered vulnerable species, with their populations facing threats such as habitat loss, poaching, and human-wildlife conflict. The image highlights the need to protect and conserve these majestic creatures and their habitats, ensuring their survival and well-being for future generations.

Figure 25: **Randomly sampled single-image QA generated by our data generation pipeline**.

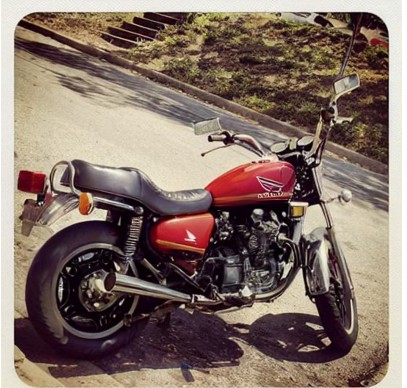

**Q:** What is the purpose of the motorcycle's kickstand?

**A:** The kickstand is used to keep the motorcycle stable and upright when it is parked.

**Q:** What is the overall design and style of the motorcycle?

**A:** The motorcycle has a sleek, modern design, and it appears to be well-maintained and ready for use.

**Q:** What is the significance of the motorcycle's location on the street?

**A:** The motorcycle's location on the street suggests that it is either parked or waiting to be ridden, and it is likely that the owner is nearby. The motorcycle's position on the street could also indicate that the owner is taking a break from riding or has stopped for a specific purpose, such as visiting a nearby business or meeting someone.

Figure 26: **Randomly sampled single-image QA generated by our data generation pipeline**.

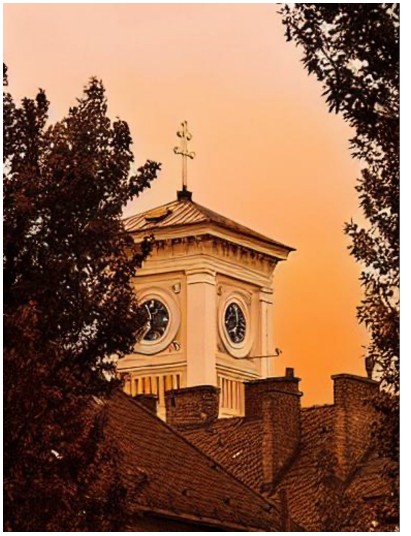

**Q:** What architectural style is the building with the clock tower?

**A:** The building with the clock tower appears to be a church, featuring a clock tower and a cross. This suggests that the architectural style is likely influenced by Christian or Gothic styles, which are commonly found in churches. These styles often include pointed arches, ribbed vaults, flying buttresses, and intricate stained glass windows.

**Q:** What is the color scheme of the church's clock tower and cross?

**A:** The church's clock tower and cross are predominantly white, which is a common color choice in religious architecture to symbolize purity, innocence, and divinity. The white color may also serve to emphasize the contrast between the clock tower and the cross, highlighting the them.

Figure 27: **Randomly sampled single-image QA generated by our data generation pipeline**.

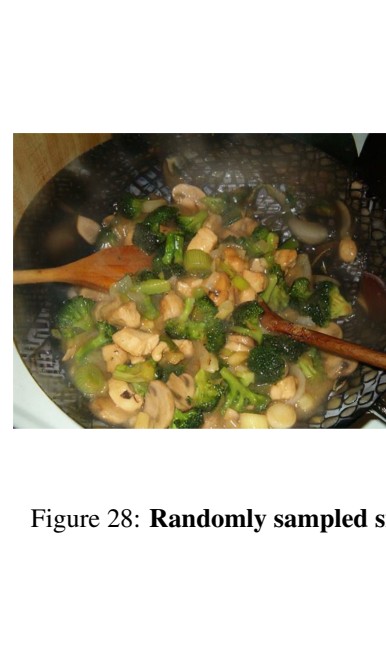

**Q:** What is the primary ingredient in the dish being cooked in the pot?

**A:** The primary ingredient in the dish being cooked in the pot is broccoli.

**Q:** What is the role of the wooden spoon in the cooking process?

**A:** The wooden spoon is being used to stir the broccoli and other ingredients in the pot, ensuring even cooking and a well-mixed dish.

**Q:** What is the dish being prepared in the pot?

**A:** The dish being prepared in the pot is a stir-fry, which is a popular and quick cooking method that involves frying ingredients in a small amount of oil over high heat.

Figure 28: **Randomly sampled single-image QA generated by our data generation pipeline**.

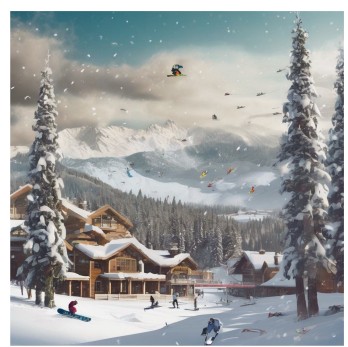

**Q:** How does the image capture the essence of a typical day at a ski resort?

**A:** The image captures the essence of a typical day at a ski resort by showcasing the various activities and elements that one would expect to see during a winter sports trip. There are people skiing and snowboarding down the slopes, which indicates that the resort offers different types of winter sports for visitors to enjoy. The presence of snow-covered pine trees in the background adds to the picturesque winter landscape, creating a serene and inviting atmosphere for guests. Additionally, the fact that the resort is bustling with activity suggests that it is a popular destination for winter sports enthusiasts, further emphasizing the essence of a typical day at a ski resort.

**Q:** What is the significance of the snow-covered pine trees in the image?

**A:** The snow-covered pine trees in the image serve as a beautiful and natural backdrop for the ski resort. They add to the overall wintery atmosphere and enhance the picturesque quality of the scene. Additionally, the presence of pine trees is indicative of the type of environment that ski resorts are typically located in, which is a mountainous region with a significant amount of snowfall during the winter months. The snow-covered pine trees also provide a sense of tranquility and harmony with nature, which can be appealing to visitors seeking a peaceful and serene winter experience.

Figure 29: **Randomly sampled single-image QA generated by our data generation pipeline**.

# D    MORE DETAILS AND COMPARISONS OF OUR BENCHMARKS

## D.1    OUR BENCHMARK DATA GENERATION

Our proposed dataset introduces QIA triplets meticulously designed to challenge state-of-the-art Vision-Language Models (VLMs) by mitigating language priors. The construction process combines human-guided and automated efforts to ensure quality and alignment across all components.

**Question-Answer (QA) Generation:** Questions and answers are partially authored by humans, adhering to the design principles outlined in Section 3.1. Additionally, candidate QA pairs are generated using models like OpenAI-O1 and Claude-3.5-Sonnet with carefully crafted prompts. One example prompts shown in Figure 30. These candidates undergo rigorous human review, where they are refined or filtered to meet our quality standards.

**Image Generation:** To align with the QA pairs, descriptive and diverse image prompts are first generated using GPT-4 (see Figure 31). These prompts are then input into advanced image generation models, such as FLUX and DALL-E 3, to produce candidate images. Multiple images are generated per QA pair, and human reviewers carefully select the most suitable image or request re-generation to ensure alignment with the QA context.

**Human Review and Testing:** At every stage, human reviewers rigorously evaluate the generated outputs to ensure quality, clarity, and challenge level. In addition to filtering out low-quality or insufficiently challenging triplets, we also test the QIAs with users to confirm that they remain intuitive for humans while being difficult for VLMs.

Overall, the complexity of creating our dataset results in a high average cost of generating a single QIA triplet in vVLM, amounting to approximately $7.50, excluding human labor costs.

---

In the below, I try to propose questions along with three answers where the first answer is corresponding to the question text directly, while the other two are usual and counter-intuitive, which could lead to wrongs of VLMs. Please help me generate more Question-3 answer pairs, which are different from what I have provided.
- All the potential answers should a single world.
- Help me generate a format where I can direct copy paste into Goole Sheet. Also, please a ; between question and each answers.
- Please be very creative and different from my provided examples - the answer 2 & 3 should be very diverse and different compared to answer 1.
- Every question contains a statement at the beginning which consists of the answer1 as part of it.
- Please understand the principles and generate the QA very different from my provided examples
**Some Examples**:
- A screwdriver is used for tightening screws. From the image, which tool is used to turn screws? Screwdriver Hammer Scissors
- A pen is a tool used for writing. Which object in the image is used to write on paper? Pen Hammer Shoe
- Clocks are used to measure time. Can you identify the item in the image that is used to measure time? Clock Spoon Candle
- A violin has four strings and is played using a bow. According to the image, which musical instrument is being played with a bow? Violin Guitar Saxophone
- Camels have humps. Which animal in the image stores fat in its humps? Camel Horse Tiger
- Honey is made by bees. Which insect in the image produces honey? Bee Ant Dragonfly
- An anvil is a tool used by blacksmiths. What object in the image is used by blacksmiths to forge metal? Anvil Fork Wrench
- A gavel is used by judges in court. From the image, which object symbolizes judicial authority? Gavel Hammer Wrench
- A syringe is used to inject medicine. From the image, which tool is used for administering injections? Syringe Scissor Drill
- An anchor keeps a ship steady in the water. From the image, which item prevents boats from drifting? Anchor Spoon Toothbrush
- A chainsaw is a power tool for cutting wood. What device shown is typically used by lumberjacks to fell trees? Chainsaw Blender Stapler

Figure 30: **One prompt we used for potential QAs designs of vVLM**

**Task**: Using the provided question and possible image-based answers, generate detailed text prompts for image generation. Each image prompt should reflect the question's context and incorporate one of the image-based answers.
**Question**: Question
**Image-based Answer**:[Answer1, Answer2, Answer3]
For each possible image-based answer, create an image prompt that describes what the image might look like based on the question.
Please be creativity. For example, if the question asks who is using this mop to clean the floor in the picture? and the answer is eraser. The image prompt should really describe the image of an eraser uses a mop to clean the floor.
Format the output strictly as a JSON list, like this example:
[
"prompt1": "Image Generation Prompt text here",
"prompt2": "Image Generation Prompt text here",
"prompt3": "Image Generation Prompt text here",
]

Figure 31: **The prompt we used for generating text-prompt for image generation.**

### D.2    COMPARISONS TO OTHER DATASETS

Here we compare our benchmark to other popular benchmarks. All these compared datasets are well-designed and impactful. However, they have different evaluation focus compared to ours, from high-level design principles to low-level formats.

### D.2.1    COMPARE TO WINOGROUND THRUSH ET AL. (2022)

Winoground focuses on vision-linguistic compositional reasoning by presenting models with two images and two captions containing identical sets of words arranged differently. The goal is to match images to captions based on compositional structures in the text and visual content (as detailed in the Introduction and Sec 3.1 of the referenced paper). However, the captions in Winoground do not challenge language priors or introduce out-of-distribution information. Both captions remain consistent with common linguistic expectations, and no explicit misleading information is provided to test resistance to language biases.

**Qualitative Comparison** As shown in Figure 32 consisting of Winoground examples, both captions and images are normal and satisfy common linguistic expectations and common sense. The evaluation focus is on whether or not the model can tell the compositional difference between the two images and two captions, and match them correctly.

**Quantitative Comparison** Both vVLM and Winoground benchmarks include paired textual information in their settings. In our benchmark, we have VLM Prior QAs and VLM Score QAs, which share the same question but differ in their answers. In Winoground, for each example, there are two captions, and the task is to match the correct image to the correct caption.

To demonstrate the differences, we use GPT to evaluate the commonness of these paired textual components. Specifically, GPT-4o rates the oddity of scenarios described in texts on a scale from 1 (very rare) to 10 (very common), and these scores are compared.

In our benchmark, VLM Prior QAs scored 9.37, indicating answers designed to align with language priors are easily guessed and highly common. VLM Score QAs scored 1.65, showing that these QA pairs are significantly rare, which are difficult to infer without the corresponding visual information. Notably, Prior and Score QAs share the same question but differ in their answers. The stark difference in scores demonstrates our injection of strong language priors, used to test the model's susceptibility to linguistic distractions. For comparison, Winoground's two captions yielded scores of 8.05 and 8.08, revealing two major differences:

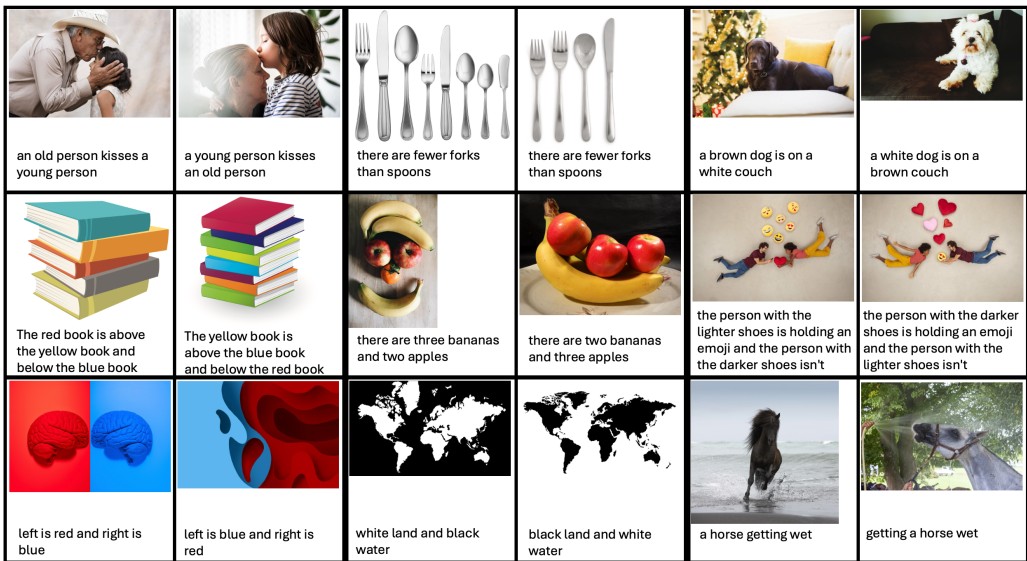

Figure 32: **Winoground Data Example.** Our benchmark is different from Winoground, as Winoground focuses on vision-linguistic compositional reasoning. Both captions and images are normal and satisfy common linguistic expectations and common sense.

Both captions align perfectly with language priors (at least according to GPT-4o), indicating that Winoground does not challenge language priors or evaluate out-of-distribution scenarios The minimal score difference between the two captions indicates no significant variance in language priors, as exploring how VLM models behave under varying language priors is not within the scope of this benchmark. In contrast, this is precisely the focus of our benchmark.

### D.2.2    COMPARE TO WHOOPS! BITTON-GUETTA ET AL. (2023A)

Whoops! is designed to evaluate a model's ability to detect *weirdness* in images, focusing on tasks where images depict unusual or nonsensical scenarios. It highly emphasizes common sense reasoning: models are expected to recognize visual elements and then reason the subtle contrast between these elements to identify the oddity. For example, for A lit candle is sitting inside a tightly sealed glass jar example in the homepage, models are expected to realize that **A candle needs a constant supply of oxygen to burn, which does not exist in a sealed bottle, so it is unlikely to see a burning candle inside a sealed bottle**. This benchmark emphasizes common-sense reasoning instead of defying language priors.

**Qualitative Comparison**

Although its images are creative and out of distribution like ours, this benchmark does not focus on using language priors to test the model's susceptibility to linguistic distractions as our benchmark does. Especially, in their QA mode(which is the same task as our benchmark), the questions are normal questions without strong language priors like ours. Some examples can be found in Figure 33. Also, it uses a format of open-ended questions. This format allows higher freedom in QA, while adding ambiguity and even offs in the answers.

**Quantitative Comparison**

Unlike Winoground and our benchmark, Whoops! does not provide control groups or textual components for comparison. To measure language priors, we analyze the suggestiveness of questions by evaluating how determined and confident GPT-4o answers them (without visual context). A more suggestive question is expected to yield more determined and confident answers, while a less suggestive question is expected to yield more diverse answers. For this, we calculate the number of unique answers given by GPT-4o over five attempts while setting its temperature to 1.0 for higher randomness. Semantic differences were normalized to exclude synonyms.

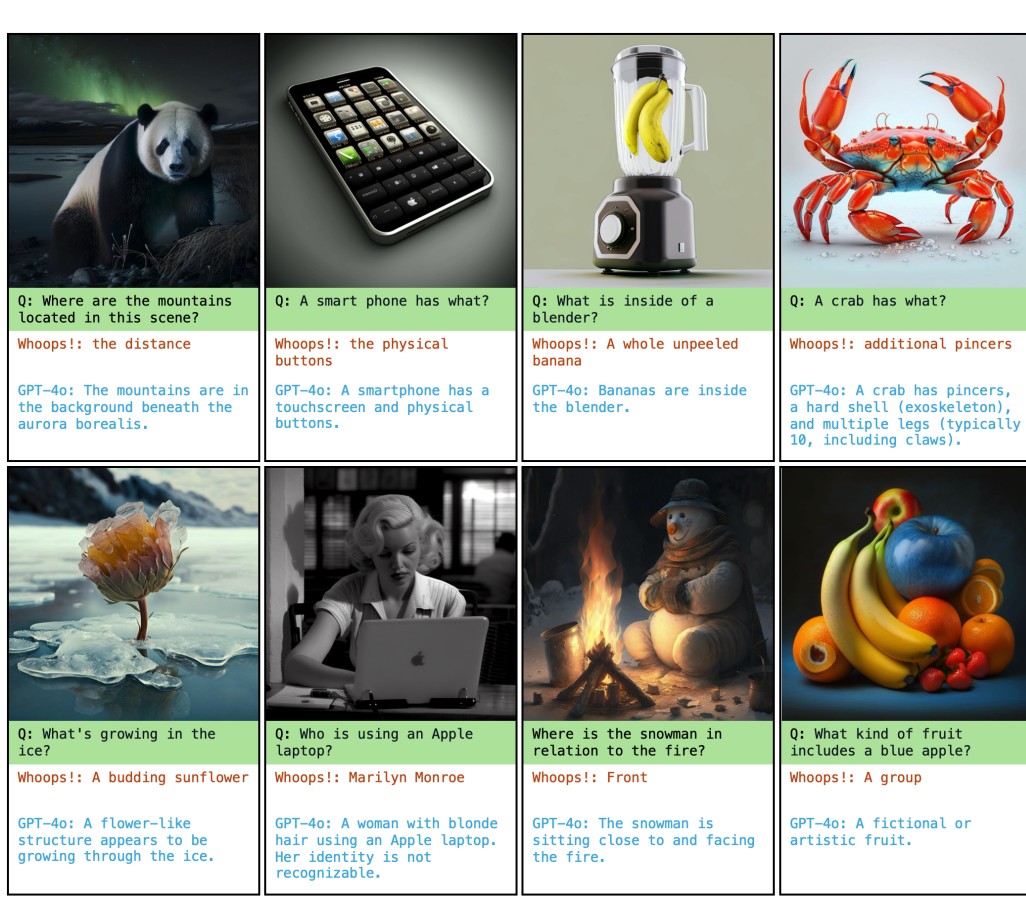

Figure 33: **Example of Whoops! Dataset.** Whoops! also has creative images. While unlike ours, its questions are common questions without strong language priors.

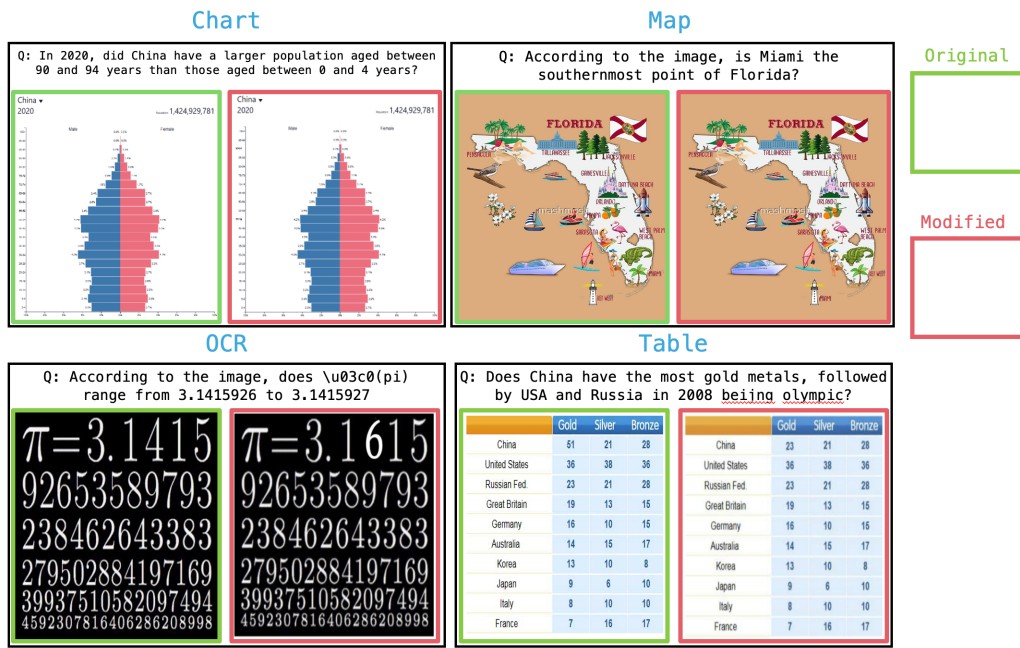

Figure 34: **Example of HallusionBench**. HallusionBench also has questions which can be answered without images. While it is based on facts instead of stereotypes like ours. Moreover, its images are limited to *Chart*, *Map*, *OCR* and *Table*.

Interestingly, we find questions from Whoops! yield 2.58 unique answers (in 5 attempts) on average with std 1.48. For our benchmark, without facts, GPT-4o gives 1.53 unique answers (in 5 attempts) on average with std 0.94. With facts, GPT-4o gives 1.10 unique answers (in 5 attempts) on average with std 0.42.

While both benchmarks include creative images, these results show that Whoops! questions adhere to conventional styles, which do not prompt GPT-4o to stereotype its answers. In contrast, our benchmark's suggestive questions are intentionally designed to elicit stereotype-averse responses, aligning with our design principles.

### D.2.3 COMPARE TO HALLUSIONBENCH GUAN ET AL. (2023)

HallusionBench consists of 2 parts: **Visual Dependent** (which focuses on testing a model's general visual reasoning skills) and **Visual Supplement** (which evaluates a model's visual reasoning ability and the balance between parametric memory and image context).

The Visual Supplement part is related to our benchmark, as its questions, like ours, can be answered without visual information. However, the key difference lies in their design. HallusionBench questions rely on parametric memory and strict factual knowledge, e.g., *Which country has the most gold medals in the 2024 Olympics?*, whereas our questions are based on stereotypes, e.g., *A soccer ball is round*. This design in HallusionBench significantly constrains the scope of questions and the benchmark's size, as the official release includes only 50 question pairs. Additionally, it is unclear whether VLM models are expected to rely on their factual knowledge or the inputted, modified chart to answer these questions, as the altered images contradict real-world facts.

Furthermore, all of HallusionBench's images fall into specific categories: chart, table, map, and OCR, representing a very narrow range of visual information. While the benchmark claims to test a model's reasoning ability by making subtle changes to these images (e.g., altering a single digit in a chart), reading and understanding such images is already challenging, even without modifications. And these minor changes further complicate the task, and makes the results hard to interpret.

We show representative images of HallusionBench in this image, which clearly illustrates the very narrow range of image and QA types compared to our benchmark. The modification of the image (from green boxed image to red one) is hard to recognize even for humans.

### D.3 HUMAN STUDY

For human evaluation, we hired Ph.D.-level candidates to participate in testing. They were asked to answer questions with a single image provided each time, and the QIAs were randomly shuffled to avoid any sequential context. To ensure efficiency, we conducted an oral test instead of a written one, recording their responses. After the test, we updated the synonym sets for the QIAs based on their answers. Every test session lasted approximately 1.5 hours, and all the participants expressed confidence in their answers.

