# OpenReview forum: "vVLM: Exploring Visual Reasoning in VLMs against Language Priors"
_ICLR.cc/2025/Conference — Submitted to ICLR 2025_

### Official Review · Reviewer_QTDS · 2024-11-03

**Soundness:** 1
**Presentation:** 2
**Contribution:** 2
**Rating:** 3
**Confidence:** 4

**Summary:**

This paper investigates the bias of Vision-Language Models (VLMs) towards language by creating a synthetic Visual Question Answering (VQA) dataset of 900 images using state-of-the-art (SOTA) image generation models DALL.E-3 and Flux. The authors demonstrate that VLMs tend to rely heavily on language cues when interpreting visual questions, which can limit their performance in truly visual reasoning tasks. To address this issue, they apply Direct Preference Optimization (DPO) training on top of VLMs, using self-generated VQA pairs and corresponding images from pre-trained image generation and editing models. This approach aims to mitigate the language bias and improve the alignment of VLMs with visual content.

**Strengths:**

1. This paper introduces an automatic way to check language bias of VLMs by synthetically generating images using SOTA image generation models.
2. This paper proposes Vision-Centered DPO by running DPO on automatically generated good/bad data where a QIA pairs, the QA remains the same and the Image is perturbed.

**Weaknesses:**

1. The vVLM benchmark is interesting but there is no comparison with other datasets (e.g., HallusionBench[a]). The paper mentions that the generated​​ images vary in texture, shape, conceptual combinations, hallucinated components, and proverb. However, the distribution of these features in the data is not specified. It would be insightful to the community to learn in which dimension the existing VLMs are failing mostly.
2. The DPO experiment and result section seems incomplete and is hard to follow through as there are little details of the training data, benchmarks and the purpose of choosing these benchmarks. It is unclear what amount of data has been synthetically generated for DPO training and what is the quality of the data. Image DPO works in some cases and not the best choice in others and there is no insight or empirical analysis to justify when Image DPO is useful.


(a) https://www.researchgate.net/profile/Fuxiao-Liu-2/publication/375595895_HALLUSIONBENCH_You_See_What_You_Think_Or_You_Think_What_You_See_An_Image-Context_Reasoning_Benchmark_Challenging_for_GPT-4Vision_LLaVA-15_and_Other_Multi-modality_Models/links/65518863b86a1d521bd839cf/HALLUSIONBENCH-You-See-What-You-Think-Or-You-Think-What-You-See-An-Image-Context-Reasoning-Benchmark-Challenging-for-GPT-4Vision-LLaVA-15-and-Other-Multi-modality-Models.pdf

**Questions:**

Q1. "Interestingly, we observe a strong correlation between the difficulty VLMs face in interpreting images and the challenges encountered by image generation models in producing those images. For example, when an image is easily generated, even weaker VLMs can correctly answer the associated QIA with relative ease." - What is the definition of difficulty here? Could you share some qualitative and quantitative analysis here?

Q2. "On vVLM$^P$-Score, the Image-DPO method achieved the second-best performance which indicates that with only question image combination, Image DPO is less effective. Is there any insight why this is the case?

Q3. In addition, in presence of prior, Image DPO is worse than RLHF-V. Is this because of the quality of the data? It would have been better to provide insights and empirical analysis on why Image DPO works on certain cases and why not in other.

Q4. It is not clear from the paper which VLMs have been used to create the DPO training data. Additionally, there has been no qualitative analysis on the quality of the generated  image and question-answer pairs.

Q5. It would be interesting to see a combination of both Image and Text-DPO and their impact on downstream tasks.

Q6. In table 3, we can see that with increasing model size the performance gap between Image DPO training and non-Image DPO training is reducing. It would be great to see any experiment that addresses the effectiveness of various DPO training with scaling the model size.

Q7. What is the performance of Image DPO on complex compositionality tasks (e.g., Winoground [1]) that checks image understanding ability of VLMs?

1. https://arxiv.org/abs/2204.03162

---

> ### Author Response · Authors · 2024-11-23
> **Author Rebuttal (1/7)**
>
> We sincerely appreciate the detailed and insightful feedback provided by Reviewer QTDS. Below, we have carefully addressed the opportunities for improvement you highlighted, particularly concerning the performance of our proposed image-DPO method. **Our response is thorough and detailed, reflecting the seriousness with which we value and engage with your thoughtful efforts: you really list very detailed potential points we could improve, thanks.**
>
> Additionally, we kindly emphasize that our paper presents two primary contributions: (1) the development of a well-established benchmark designed to examine language prior biases in VLMs, and (2) the introduction of Image-DPO aimed at enhancing VLMs’ focus on visual inputs. **Particularly, our selected primary area is datasets and benchmarks, where we feel a bit you under value our proposed benchmark.** We hope that you will evaluate our work holistically, considering the comprehensive improvements we have made which addressed the identified weaknesses of Image-DPO.
>
> ---
>
> > The vVLM benchmark is interesting but there is no comparison with other datasets (e.g., HallusionBench[a]). The paper mentions that the generated​​ images vary in texture, shape, conceptual combinations, hallucinated components, and proverb. However, the distribution of these features in the data is not specified. It would be insightful to the community to learn in which dimension the existing VLMs are failing mostly.
>
> We greatly appreciate the valuable feedback! During the rebuttal, **we performed comparisons with Winoground ((Thrush 2022 et al.), Whoop! (Bitton-Guetta 2023 et al.), and HallusionBench (Guan 2024 et al.)**. Also, we added the categorical information and related discussion in Section 3.2. We list the detailed information in the below.
>
> # Compare with other Datasets
>
> ## Winoground (Thrush 2022 et al.)
>
> Winoground focuses on vision-linguistic compositional reasoning by presenting models with two images and two captions containing identical sets of words arranged differently. The goal is to match images to captions based on compositional structures in the text and visual content (as detailed in the Introduction and Sec 3.1 of the referenced paper). However, the captions in Winoground do not challenge language priors or introduce out-of-distribution information. Both captions remain consistent with common linguistic expectations, and no explicit misleading information is provided to test resistance to language biases.
>
> ### Qualitative Comparison
> **As shown in the provided [figure](https://imgdrop.io/image/grK9D) consisting of Winoground examples**, both captions and images are normal and satisfy common linguistic expectations and common sense. The evaluation focus is on whether or not the model can tell the compositional difference between the two images & two captions, and match them correctly.
>
> ### Quantitative Comparison
>
> Both vVLM and Winoground benchmarks include paired textual information in their settings.
> In our benchmark, we have **VLM Prior QAs** and **VLM Score QAs**, which share the same question but differ in their answers.
> In Winoground, for each example, there are two captions, and the task is to match the correct image to the correct caption.
>
> To demonstrate the differences, we use GPT to evaluate the commonness of these paired textual components. Specifically, GPT-4o rates the oddity of scenarios described in texts on a scale from 1 (very rare) to 10 (very common), and these scores are compared.
>
> In our benchmark,
> - **VLM Prior QAs** scored **9.37**, indicating answers designed to align with language priors are easily guessed and highly common.
> - **VLM Score QAs** scored **1.65**, showing that these QA pairs are significantly rare, which are difficult to infer without the corresponding visual information.
> - Notably, Prior and Score QAs share the same question but differ in their answers.
> The stark difference in scores demonstrates our injection of strong language priors, used to test the model's susceptibility to linguistic distractions.
>
> For comparison, Winoground's two captions yielded scores of **8.05** and **8.08**, revealing two major differences:
> - Both captions align perfectly with language priors (at least according to GPT-4o), indicating that Winoground **does not** challenge language priors or evaluate out-of-distribution scenarios
> - The minimal score difference between the two captions indicates no significant variance in language priors, as exploring how VLM models behave under varying language priors is not within the scope of this benchmark. In contrast, this is precisely the focus of our benchmark.

---

> ### Author Response · Authors · 2024-11-23
> **Author Rebuttal (2/7)**
>
> ## Whoop! (Bitton-Guetta 2023 et al.)
>
> Whoops! is designed to evaluate a model's ability to detect "weirdness" in images, focusing on tasks where images depict unusual or nonsensical scenarios.
> It highly emphasizes common sense reasoning: models are expected to recognize visual elements and then reason the subtle contrast between these elements to identify the oddity.
> For example, for `A lit candle is sitting inside a tightly sealed glass jar` example in the [homepage](https://whoops-benchmark.github.io/), model is expected to realize that *A candle needs a constant supply of oxygen to burn, which does not exist in a sealed bottle, so it is unlikely to see a burning candle inside a sealed bottle*. This benchmark emphasizes common-sense reasoning instead of defying language priors.
>
> ### Qualitative Comparison
> Although its images are creative and out of distribution like ours, this benchmark does not utilize language priors to test the model's susceptibility to linguistic distractions as our benchmark does.
> Especially, in their QA mode(which is the same task as our benchmark), the questions are all very normal questions without any strong language priors. **Some examples can be found at [figure](https://imgdrop.io/image/gRcat).**
> Also, it uses a format of open-ended questions, which is hard to evaluate and contains ambiguities and even errors.
>
> ### Quantitative Comparison
> Unlike Winoground and our benchmark, Whoops! does not provide control groups or textual components for comparison.
> To measure language priors, we analyze the suggestiveness of questions by evaluating how determined and confident GPT-4o answers them (without visual context).
> A more suggestive question is expected to yield more determined and confident answers, while a less suggestive question is expected to yield more diverse answers.
> For this, we calculate the number of unique answers given by GPT-4o over five attempts while setting its temperature to 1.0 for higher randomness. Semantic differences were normalized to exclude synonyms.
>
> The results reveal:
>
> - Questions from Whoops! yielded **2.58** unique answers (on average) with a standard deviation of **1.48**.
>
> - In our benchmark:
>     - Without facts, GPT-4o gave **1.53** unique answers (std **0.94**).
>     - With facts, GPT-4o gave **1.10** unique answers (std **0.42**).
>
> Interestingly, we find questions from Whoops! yield **2.58** unique answers (in 5 attempts) on average with std **1.48**.
> For our benchmark, without facts, GPT-4o gives 1.53 unique answers (in 5 attempts) on average with std 0.94.
> With facts, GPT-4o gives 1.10 unique answers (in 5 attempts) on average with std 0.42.
>
> While both benchmarks include creative images, these results show that Whoops! questions adhere to conventional styles, which do not prompt GPT-4o to stereotype its answers.
> In contrast, our benchmark's suggestive questions are intentionally designed to elicit stereotype-averse responses, aligning with our design principles.

---

> ### Author Response · Authors · 2024-11-23
> **Author Rebuttal (3/7)**
>
> ## HallusionBench (Guan 2024 et al.)
>
> HallusionBench consists of 2 parts: **Visual Dependent** (which focuses on testing a model's general visual reasoning skills) and **Visual Supplement** (which evaluates a model's visual reasoning ability and the balance between parametric memory and image context).
>
> The **Visual Supplement** part is related to our benchmark, as its questions, like ours, can be answered without visual information. However, the key difference lies in their design. HallusionBench questions rely on **parametric memory** and strict factual knowledge, e.g., *Which country has the most gold medals in the 2024 Olympics?*, whereas our questions are based on stereotypes, e.g., *A soccer ball is round*. This design in HallusionBench significantly constrains the scope of questions and the benchmark's size, as the official release includes only 50 question pairs.
> Additionally, it is unclear whether VLM models are expected to rely on their factual knowledge or the inputted, modified chart to answer these questions, as the altered images contradict real-world facts.
>
> Furthermore, all of HallusionBench's images fall into specific categories: **chart**, **table**, **map**, and **OCR**, representing a very narrow range of visual information. While the benchmark claims to test a model's reasoning ability by making subtle changes to these images (e.g., altering a single digit in a chart), reading and understanding such images is already challenging, even without modifications. And these minor changes further complicate the task, and makes the results hard to interpret.
>
> **We show representative images of HallusionBench in [this image](https://imgdrop.io/image/g1BJU)**, which clearly illustrates the very narrow range of image and QA types compared to our benchmark. The modification of the image (from green boxed image to red one)  is hard to recognize even for humans.
>
> # Categorical Information
>
> Thank you for raising this insightful point! Initially, we did not provide detailed discussions on specific categories because most of our data samples are not confined to a single category. Instead, they often incorporate multiple variations—such as shapes and conceptual combinations, as in the moon example (e.g., “round moon + hexagon” versus “hexagonal moon”)—designed to challenge even the most advanced VLMs. However, upon your suggestion, we recognized the importance of addressing this in our main paper.
>
> In our rebuttal, we addressed this concern by categorizing each sample, including those that span multiple categories, and presenting this information clearly for improved clarity. We also expanded Section 3.2 to include detailed categorical statistics and related discussions. Additionally, we discussed the relevant categories for the typical examples featured in the teaser to provide better context.

---

> ### Author Response · Authors · 2024-11-23
> **Author Rebuttal (4/7)**
>
> > The DPO experiment and result section seems incomplete and is hard to follow through as there are little details of the training data, benchmarks and the purpose of choosing these benchmarks. It is unclear what amount of data has been synthetically generated for DPO training and what is the quality of the data. Image DPO works in some cases and not the best choice in others and there is no insight or empirical analysis to justify when Image DPO is useful.
>
> We sincerely thank you for sharing your concerns! We addressed your concerns in the rebuttal:
>
>
> - We initially included the experimental details in Appendix C.1, but we have now revised and clarified the content. Please refer to the updated Appendix C.2 for more details.
> - Upon your suggestion, we also revised our main paper to include better explanations around the details of training and the choice of benchmarks. We’ve selected the benchmarks and updated the table accordingly, examining visual reasoning against language prior, compositionality, conversation, and hallucination.
> - During the rebuttal, we conducted additional experiments to demonstrate that ImageDPO becomes increasingly effective with scaling synthetic data. We will provide the detailed results in our subsequent responses.
> - Furthermore, we have added a theoretical justification for ImageDPO, showing that our objective effectively optimizes the upper bound of the RLHF objective.  Please refer to Appendix B for the detailed math proofs.
>
>
> ---
>
> > Q1. "Interestingly, we observe a strong correlation between the difficulty VLMs face in interpreting images and the challenges encountered by image generation models in producing those images. For example, when an image is easily generated, even weaker VLMs can correctly answer the associated QIA with relative ease." - What is the definition of difficulty here? Could you share some qualitative and quantitative analysis here?
>
> To clarify, we face two challenges here:
> - Image Generation Difficulty: This refers to the number of attempts required to generate an image that adequately satisfies the given text prompt using an image generation model.
> - VLM Difficulty: This pertains to the challenge posed to VLMs, as certain “weaker” models, such as LLaVA-7B and Molmo-1B, are able to answer the image-questions correctly.
>
> We are confident in our conclusion that images with greater generation complexity posed more significant challenges for VLMs. Although we did not maintain quantitative records to specifically track generation metrics for each image, we provided qualitative analyses during the rebuttal phase that underscore the correlation between image generation complexity and VLM performance.
>
> **Please refer to [image](https://imgdrop.io/image/gcf3I) for qualitative results**, where we present multiple image pairs for analysis. In each pair, the left-hand image was relatively easy to generate as they are our prior answers which satisfy language priors. Similarly, the right-hand image in the first row was also straightforward to generate. For example, tasks like changing the color of a bridge to yellow, adding wings to a sloth, or making a koala jump required only one or two trials to produce satisfactory results.
>
> In contrast, the right-hand images in the second row represent more complex scenarios. For instance:
> - In the second row, second column, generating an image of a man riding a seal was straightforward. However, adding additional elements—such as placing the scene on a yellow grass field and incorporating horses in the background—posed significant challenges.
> - In the second row, fourth column, creating an image of a person on a raft was simple, but adding a nearby gondola and ensuring it was empty proved far more difficult.
> - In the second row, sixth column, generating a honeycomb with cube or square shapes presented unique challenges, as honeycombs are predominantly hexagonal. (Note: both cube/square and hexagon/hexagonal answers were accepted as correct in our evaluation pipeline.)
>
> These challenging images are often required over multiple iterations (> 20) to produce a satisfactory result. Overall, VLMs performed well on images that were easy to generate, even when using less powerful models like LLaVA-7B. However, their performance consistently degraded for tasks requiring significant iterative refinement. This limitation was observed across both open-source models and commercial ones, including GPT-4O and Claude-Sonnet-3.5.

---

> ### Author Response · Authors · 2024-11-23
> **Author Rebuttal (5/7)**
>
> > Q2. "On vVLM-Score, the Image-DPO method achieved the second-best performance which indicates that with only question image combination, Image DPO is less effective. Is there any insight why this is the case?
>
> > Q3. In addition, in presence of prior, Image DPO is worse than RLHF-V. Is this because of the quality of the data? It would have been better to provide insights and empirical analysis on why Image DPO works on certain cases and why not in other.
>
> Thank you for this excellent question! We scale the data and training steps used for image DPO training and present the results below, which demonstrate a substantial improvement in performance.
>
> DPO optimizes a KL divergence objective, which is relatively less efficient and requires more training data and steps to converge. Scaling the data and training steps effectively enhances performance, as shown in our results. Notably, all the new data are generated by our proposed pipeline without introducing any external datasets.
>
> It is a valuable observation that image DPO appears to be more effective for QAs involving facts ($vVLM^{F}$), as the absolute performance improvement is more significant. This outcome is expected and exactly aligns with our design hypothesis for Image DPO.
>
> The failure of the benchmark can roughly be attributed to two main factors: the model's inherent ability limitations (e.g., model architecture, training data, etc.) and the influence of language priors in our benchmark. By design, $vVLM^{F}$ shares the exact same QIA as $vVLM^{P}$, except for the inclusion of additional misleading facts. The improvement difference between $vVLM^{F}$ and $vVLM^{P}$ reflects the model's ability to resist language priors, as the only difference between them is the additional misleading facts. Visual reasoning ability improvements should be reflected in both $vVLM^{F}$ and $vVLM^{P}$.
>
> The greater improvement difference between $vVLM^{F}$  and  $vVLM^{P}$ suggests that the Image DPO-trained model is less easily misled by language priors and focuses more on visual information to provide correct answers. Its visual reasoning ability is still constrained by its fundamental architecture and training data, although improved by finetuning pipelines.
>
> Other baselines, such as LLaVA-1.5-7B + CSR, show significant improvement on $vVLM^{P}$ but perform even worse on $vVLM^{F}$, indicating a higher susceptibility to being misled by language priors. This actually reflects that these methods  are useful to improve VLMs performance while not helpful to teach models to focus on visual information, which is stated in our Image DPO motivation.
>
>
>
> | Model                       |$vVLM^F_{score}$ | $vVLM^F_{prior}$ | $vVLM^P_{score}$ | $vVLM^P_{prior}$ |
> |-----------------------------|------------------|------------------|------------------|------------------|
> | LLaVA-1.5-7B               | 29.67           | 71.33           | 37.67           | 65.67           |
> | LLaVA-1.5-7B + CSR         | 27.50           | 64.33           | 40.84       | 63.67           |
> | LLaVA-1.5-7B + RLHF-V      | 29.50           | 75.00           | 36.33           | 65.33           |
> | LLaVA-1.5-7B + Text-DPO    | 31.34           | 71.67           | 37.83           | 65.67           |
> | LLaVA-1.5-7B + Image-DPO | **34.17**       | **77.67**       | *38.67*         | **67.33**       |

---

> ### Author Response · Authors · 2024-11-23
> **Author Rebuttal (6/7)**
>
> > Q4. It is not clear from the paper which VLMs have been used to create the DPO training data. Additionally, there has been no qualitative analysis on the quality of the generated image and question-answer pairs.
>
> In the submitted version, we may have included too many details in the supplementary material while leaving limited information in the main paper due to space constraints. We will address this in the revision after further discussion with you.
>
> Regarding the models, the DPO training data for each model was generated using the model itself. For example, the LLaVA-7B model was used to generate the DPO training data for the LLaVA-7B model, and the Cambrian-8B model was used to generate the DPO training data for the Cambrian-8B model. Additionally, we have provided the text prompts used for data generation in Appendix C.1 Figure 13 and included randomly sampled generated data at Appendix C Figure 15~29.
>
>
> ---
>
> > Q5. It would be interesting to see a combination of both Image and Text-DPO and their impact on downstream tasks
>
>
>
>
> Thank you for this insightful suggestion. During the rebuttal, we conducted an ablation study combining image and text DPO for training the model. The results, shown below, indicate that this fusion-approach does not provide additional improvements on our proposed benchmark. However, we guess there could be a way to meaningfully combine the two approaches and leave it for future work.
>
> | Model                                     | $vVLM^F_{score}$ | $vVLM^P_{score}$ | MMVET | SEED  | $Chair^S$ ⬇️ | $Chair^I$ ⬇️ |
> |-------------------------------------------|----------|----------|-------|-------|--------------|--------------|
> | LLaVA-1.5-7B                             | 29.67    | 37.67    | 31.1  | 58.6  | 49.1         | 14.8         |
> | LLaVA-1.5-7B + Image-DPO + Text-DPO      | 31       | 38.33    | 30.9  | 59.32 | 49.4         | 13.4         |
> | LLaVA-1.5-7B + Image-DPO                 | 34.17    | 38.67 (not 78.67)    | 33.8  | 59.65 | 45.4         | 12.4         |
> | LLaVA-1.5-13B                            | 35.33    | 41       | 36.1  | 61.9  | 48.3         | 14.1         |
> | LLaVA-1.5-13B + Image-DPO + Text-DPO     | 37       | 41       | 37.2  | 62.04 | 45.7         | 12.4         |
> | LLaVA-1.5-13B + Image-DPO                | 38.17    | 42.5     | 37.5  | 62.31 | 42.6         | 11.6         |

---

> ### Author Response · Authors · 2024-11-23
> **Author Rebuttal (7/7)**
>
> > Q6. In table 3, we can see that with increasing model size the performance gap between Image DPO training and non-Image DPO training is reducing. It would be great to see any experiment that addresses the effectiveness of various DPO training with scaling the model size.
>
> This is an excellent point! During the rebuttal, we recognized that scaling our dataset significantly impacts performance. Additionally, as detailed in Appendix C, our proposed data pipeline allows us to generate an arbitrary amount of data. For your convenience, I’ve uploaded the [pipeline illustration figure here](https://imgdrop.io/image/overview.g1LNV).
>
> To evaluate the effect of scaling, we increased our dataset size from 80k to 800k. Furthermore, we incorporated NaturalBench (Li et al. 2024), a benchmark evaluating against natural adversarial samples suggested by Reviewer M3UT, to evaluate the aspects around compositionality and biases. The results, shown below, demonstrate that the LLava-13B model achieved further improvements with scaled data, highlighting data scaling as an effective approach to address increases in model size.
>
> | Model                                | $vVLM^F_{score}$ | $vVLM^P_{score}$ | NaturalBench-Question | NaturalBench-Image | NaturalBench-Group | MMVET | SEED | $Chair^S$ ⬇️ | $Chair^I$ ⬇️ |
> |--------------------------------------|----------|----------|------------------------|--------------------|--------------------|-------|------|--------------|--------------|
> | LLaVA-1.5-13B                        | 35.33    | 41       | 39.6                  | 44.6              | 14.8              | 36.1  | 61.9 | 48.3         | 14.1         |
> | LLaVA-1.5-13B + 80k Image-DPO        | 36.17    | 41.5     | 41.24                 | 45.74             | 16.16             | 37.2  | 62.2 | 44.7         | 12.2         |
> | LLaVA-1.5-13B + 800k Image-DPO       | 38.17    | 42.5     | 42.68                 | 47.37             | 17.16             | 37.5  | 62.31| 42.6         | 11.6         |
>
> ---
>
> > Q7. What is the performance of Image DPO on complex compositionality tasks (e.g., Winoground [1]) that checks image understanding ability of VLMs?
>
>
> Thank you for your valuable suggestion. Following your recommendation, we have conducted the evaluation, as presented below.
>
>
> | Model                          | LLaVA-1.5-7B | LLaVA-1.5-7B + Image-DPO | LLaVA-1.5-13B | LLaVA-1.5-13B + Image-DPO |
> |--------------------------------|--------------|--------------------------|---------------|---------------------------|
> | Winoground                     | 20           | 22.75                    | 29.25         | 31.5                      |
>
>
> Due to computational constraints, we are still in the process of finalizing the Image-DPO fine-tuning for the Cambrian model. We expect to complete this process before the conclusion of the discussion and update the results once we got the new ckpt.

---

### Official Review · Reviewer_M3UT · 2024-11-04

**Soundness:** 3
**Presentation:** 3
**Contribution:** 2
**Rating:** 6
**Confidence:** 4

**Summary:**

This paper discusses the issue of language biases in Vision Language Models (VLMs) and introduces a benchmark that prioritizes visual reasoning in visual question answering (VQA). Additionally, it uses good-bad image pairs for DPO training, leading to improved performance on models like LLaVA-v1.5 and Cambrian.

**Strengths:**

This paper addresses a significant and currently trending issue in VLMs: language bias. It proposes a scalable data generation pipeline, and the introduced DPO training method also appears to be intriguing. The writing of the paper is fluent and the sections are clearly structured.

**Weaknesses:**

The examples presented in the paper appear somewhat confusing, such as the image corresponding to the question "In what city does the image depict the Statue of Liberty?" and the image for "What is the largest long-necked animal shown in the image?" These examples seem counterintuitive and are even not easy for humans to answer correctly. Additionally, the paper lacks a discussion on the quality control of the dataset.

**Questions:**

1. How is the quality of the dataset ensured during the generation process to avoid discriminate, incorrect images and the awkward and meaningless questions? Are the dataset and code open source?

2. The abstract claims that “our dataset consists of visually intricate images that vary in texture, shape, conceptual combinations, hallucinated components, and proverbs,” but the paper lacks discussion on different categories.

3. In the experiments, it might be better to include experiments on benchmarks that eliminate bias, such as [NaturalBench](https://arxiv.org/pdf/2410.14669?) .

---

> ### Author Response · Authors · 2024-11-23
> **Author Rebuttal (1/2)**
>
> We greatly appreciate the positive review and thoughtful feedback provided by Reviewer M3UT. Below, we provide our detailed responses to your comments and incorporate the opportunities for improvement that you highlighted. Also, we have carefully addressed your ethical concerns in the response to the corresponding question below. **After reviewing our response, we kindly ask if you could remove the ethical concerns flag, or let us know if you still have any remaining ethical concerns.**
>
>
> ---
>
> > Q: The examples presented in the paper appear somewhat confusing, such as the image corresponding to the question "In what city does the image depict the Statue of Liberty?" and the image for "What is the largest long-necked animal shown in the image?" These examples seem counterintuitive and are even not easy for humans to answer correctly. Additionally, the paper lacks a discussion on the quality control of the dataset.
>
> The question-image-answer pairs in **our proposed dataset are intentionally designed to counteract language priors and present scenarios that may appear counterintuitive to common sense**. This approach enables a robust evaluation of modern VLMs, including leading commercial models like GPT-4O and Claude-Sonnet-3.5. By challenging models in this manner, we create a benchmark that can better assess their true visual reasoning capabilities.
>
> In Table 1, we report results from human testers, who achieved high accuracy rates (98.33% and 98.67%) for out-of-distribution answers, demonstrating that while our benchmark is challenging, it remains solvable by humans. This underscores the rigor of our dataset in testing VLM performance.
>
> Our benchmark consists of a carefully curated set of 300 data points, comprising a total of 900 question-image-answer pairs. To ensure the highest data quality, all authors rigorously reviewed the dataset themselves, i.e. human verifications. Also, we collected feedback from human participants to further improve the quality in terms of question description accuracy and image quality. We appreciate your suggestion and will include a more detailed discussion of the dataset’s quality control measures in our revision. Thank you for highlighting this important aspect!

---

> ### Author Response · Authors · 2024-11-23
> **Author Rebuttal (2/2)**
>
> > Q: How is the quality of the dataset ensured during the generation process to avoid discriminate, incorrect images, and awkward and meaningless questions? Are the dataset and code open source?
>
> Yes! The dataset and code will be open-sourced. To uphold high-quality and ethical standards, our team meticulously **reviewed every 300 data points at least five times, ensuring that all synthetic images and Q&A pairs comply with principles of fairness, non-discrimination, and bias prevention.**
>
> As mentioned above, our authors meticulously validated every data to eliminate any potentially discriminatory or incorrect images. Awkward or meaningless questions were also removed during this process. To further enhance clarity and accuracy, we engaged human evaluators to confirm that the answers and associated images were unambiguous and free from ethical concerns, including issues related to discrimination, bias, or fairness.
>
> For synthetic images depicting vivid human faces, we applied an additional layer of mosaic blurring to anonymize fake faces. An example of this can be seen in the last row, third image of Figure 1.
>
> We also list more data samples in Appendix A and hope they can alleviate your concerns.
>
> ---
>
> > Q: The abstract claims that “our dataset consists of visually intricate images that vary in texture, shape, conceptual combinations, hallucinated components, and proverbs,” but the paper lacks discussion on different categories.
>
> Thank you for raising this insightful point! Initially, we did not provide detailed discussions on specific categories because most of our data samples are not confined to a single category. Instead, they often combine multiple variations—such as shapes and conceptual combinations in the moon case (round moon + hexagon versus hexagonal moon) —designed to challenge even the strongest VLMs. However, upon your request, we realized we should discuss the related information in our main paper.
>
> In our rebuttal, we addressed this concern by categorizing each sample, including those spanning multiple categories, and clearly presenting this information for better clarity. Additionally, **we expanded Section 3.2 to provide detailed categorical statistics and related discussion. We also highlighted the relevant categories for the typical examples featured in the teaser.**
>
> ---
>
> > Q: In the experiments, it might be better to include experiments on benchmarks that eliminate bias, such as NaturalBench.
>
> During the rebuttal, we incorporated your suggestions and conducted ablation studies on NaturalBench to eliminate possible bias. The results are shown below, which indicate Image-DPO generally brings improvements on LLaVA-7B and 13B models.
>
> | Model                          | NaturalBench-Question | NaturalBench-Image | NaturalBench-Binary | NaturalBench-Group |
> |--------------------------------|------------|------------|------------|------------|
> | LLaVA-1.5-7B                  | 37.7       | 43.8       | 67.3       | 12.7       |
> | LLaVA-1.5-7B + Image-DPO      | 40.68 |	46.29 |	68.84 |	14.95 |
> | LLaVA-1.5-13B                 | 39.6       | 44.6       | 68.9       | 14.8       |
> | LLaVA-1.5-13B + Image-DPO     | 42.68      | 47.37      | 70.36      | 17.16      |
>
> We have incorporated this benchmark into our main table and added its citation in the revised version.
>
> Due to computational constraints, we are currently finalizing the Image-DPO fine-tuning for the Cambrian model. We anticipate completing this process before the discussion concludes. Once finalized, we will update the corresponding table.

---

### Official Review · Reviewer_Nay6 · 2024-11-05

**Soundness:** 2
**Presentation:** 2
**Contribution:** 2
**Rating:** 3
**Confidence:** 3

**Summary:**

This paper introduces a VQA benchmark that aims to ask visual questions that challenge the linguistic biases of vision–language models. This benchmark contains 300 questions with 3 image–answer pairs each. The questions are first generated by humans using OpenAI o1 and Claude-3.5-Sonnet. The three (single-word) answers are then written, such that the first $a_\text{prior}$ is likely and the latter two are unlikely given the question alone. An image is then synthetically generated for each question–answer pair using DALL-E 3 and Flux. Finally, a distractor fact is also included that reinforces the language prior in $a_\text{prior}$.

Many open- and closed- vision–language models are tested on this benchmark and show poor performance (< 70%) on the "unlikely" test examples. Human (98%) and text-only (GPT-4o; 0%) baselines are also provided.

This paper proposes a method for generating more VQA training examples (seeded from COCO, Text2VQA, Visual Genome) that are used for fine-tuning the models using DPO-based methods (the images are perturbed with pixel-level transformations to create negative examples). This method improves performance on the proposed benchmark by 1-3%.

**Strengths:**

I believe the examples in this benchmark are very creative! The synthetic nature of this data permits very interesting and challenging combinations of concepts: e.g. houses built from pitayas or zebras with checkerboard patterns (Figure 1).

I appreciate that the benchmark is evaluated on a large set of vision–language models (Table 1). The benchmark is also sufficiently challenging.

The writing in this paper is reasonably clear.

**Weaknesses:**

I have a few criticisms of this work. Unfortunately, I believe these are fundamental issues, but I would sincerely appreciate it if the authors could comment further.

1. While this work suggests that existing VQA benchmarks do not sufficiently "defy language priors", Winoground (Thrush 2022), (Hessel 2023), and Whoops! (Bitton-Guetta 2023) are vision–language benchmarks that do have this focus. Moreover, I believe these benchmarks remain sufficiently challenging, e.g. GPT-4v is SOTA and scores 60% on Winoground (Wu 2023).

2. I am quite concerned by the quality and content of the generated images. For example, in Figure 3: there is a flamingo present (alongside the tiger) in the image where "tiger" is the correct answer (and "flamingo" is supposed to be $a_\text{prior}$). Likewise, the "round" moon is actually boxed in a hexagon. Is the "horse with humps" sufficiently different from a camel? Horses don't have humps: is it fair to say that image is indeed a "horse"? Is the image of Paris with the Red Square really "Paris", or could it be "Moscow" with the Eiffel Tower? And so forth. I have looked at many examples and feel this issue is pervasive in this dataset.

3. The "Image DPO" method is not clearly motivated by the topic of "linguistic bias" that is explored in this paper. Why are the perturbations pixel-level and not linguistic? And, I am not sure that this benchmark demonstrates a particular need for more VQA training data. Altogether, the gains (1-3%) from the introduced methods are relatively small compared to the complex system involved (e.g. including image-editing and fine-tuning).

References:
- Winoground: Probing vision and language models for visio-linguistic compositionality. (Thrush, CVPR 2022)
- Do Androids Laugh at Electric Sheep? Humor “Understanding” Benchmarks from The New Yorker Caption Contest (Hessel, ACL 2023)
- Breaking Common Sense: WHOOPS! A Vision-and-Language Benchmark of Synthetic and Compositional Images (Guetta-Bitton, ICCV 2023)
- The Role of Chain-of-Thought in Complex Vision-Language Reasoning Task (Wu, 2023)

**Questions:**

Could the authors please elaborate on how they constructed their dataset? In particular, it is not obvious how questions, answers, and images are specifically generated. E.g. what prompts are used for generating questions and images? I believe answers are entirely written by humans: is that correct? These are really important details that are unclear from my reading.

I am not sure that Sec. 3.1 needs to rely on mathematical formulations and the three objectives. I think this actually makes reading the following sections a bit confusing. Could the authors please clarify how these objectives specifically relate to the construction of the questions/answers/images (if at all)? E.g. Sec. 3.2 just says "we designed answers [for/against] language priors" but does not relate this to a particular objective.

---

> ### Author Response · Authors · 2024-11-23
> **Author Rebuttal (1/6)**
>
> We greatly appreciate the detailed review and thoughtful feedback provided by Reviewer Nay6! We have carefully addressed your concerns in the following sections and incorporated the suggested improvements into our revised manuscript.
>
> However, we would like to kindly clarify that the issues you mentioned, while important, do not strike us as fundamental. **We believe these misunderstandings can be addressed during the rebuttal. Also, most of the updates do not affect our main paper.** Despite potential differences in perspective, we remain confident in the value of our proposed dataset and optimization method and appreciate your feedback in helping us improve further! If you have further concerns or find limited value in our work after reviewing our rebuttal, please feel free to share your thoughts.
>
> ---
>
> > Q: While this work suggests that existing VQA benchmarks do not sufficiently "defy language priors", Winoground (Thrush 2022), (Hessel 2023), and Whoops! (Bitton-Guetta 2023) are vision–language benchmarks that do have this focus. Moreover, I believe these benchmarks remain sufficiently challenging, e.g. GPT-4v is SOTA and scores 60% on Winoground (Wu 2023).
>
>
> **We respectfully contend that there might be a misunderstanding regarding the purpose and scope of our benchmark.** Our benchmark is fundamentally different from Winoground and Whoops! from high-level design including (**design principles**, **evaluation focus**) to low-level details (**benchmark format**).
>
> Both Winoground and Whoops! primarily evaluate the compositional reasoning ability of VQA models. While our benchmark also requires strong compositional reasoning—a necessary component for evaluating vision-language models (VLMs)—its primary aim is to explore model behavior against strong language priors, an unexplored area by the aforementioned datasets.We achieve this by carefully designing questions and answers out of the language priors, and even explicitly providing misleading information as strong language priors to distract the model's attention, which is shown in our design principle in Sec 3.1.
>
> ## Comparison with Winoground
> Winoground focuses on vision-linguistic compositional reasoning by presenting models with two images and two captions containing identical sets of words arranged differently. The goal is to match images to captions based on compositional structures in the text and visual content (as detailed in the Introduction and Sec 3.1 of the referenced paper). However, the captions in Winoground do not challenge language priors or introduce out-of-distribution information. Both captions remain consistent with common linguistic expectations, and no explicit misleading information is provided to test resistance to language biases.
>
> ### Qualitative Comparison
> **As shown in the provided [figure](https://imgdrop.io/image/grK9D) consisting of Winoground examples**, both captions and images are normal and satisfy common linguistic expectations and common sense. The evaluation focus is on whether or not the model can tell the compositional difference between the two images & two captions, and match them correctly.
>
> ### Quantitative Comparison
>
> Both vVLM and Winoground benchmarks include paired textual information in their settings.
> In our benchmark, we have **VLM Prior QAs** and **VLM Score QAs**, which share the same question but differ in their answers.
> In Winoground, for each example, there are two captions, and the task is to match the correct image to the correct caption.
>
> To demonstrate the differences, we use GPT to evaluate the commonness of these paired textual components. Specifically, GPT-4o rates the oddity of scenarios described in texts on a scale from 1 (very rare) to 10 (very common), and these scores are compared.
>
> In our benchmark,
> - **VLM Prior QAs** scored **9.37**, indicating answers designed to align with language priors are easily guessed and highly common.
> - **VLM Score QAs** scored **1.65**, showing that these QA pairs are significantly rare, which are difficult to infer without the corresponding visual information.
> - Notably, Prior and Score QAs share the same question but differ in their answers.
> The stark difference in scores demonstrates our injection of strong language priors, used to test the model's susceptibility to linguistic distractions.
>
> For comparison, Winoground's two captions yielded scores of **8.05** and **8.08**, revealing two major differences:
> - Both captions align perfectly with language priors (at least according to GPT-4o), indicating that Winoground **does not** challenge language priors or evaluate out-of-distribution scenarios
> - The minimal score difference between the two captions indicates no significant variance in language priors, as exploring how VLM models behave under varying language priors is not within the scope of this benchmark. In contrast, this is precisely the focus of our benchmark.

---

> ### Author Response · Authors · 2024-11-23
> **Author Rebuttal (2/6)**
>
> ## Comparison with Whoops!
> Whoops! is designed to evaluate a model's ability to detect "weirdness" in images, focusing on tasks where images depict unusual or nonsensical scenarios.
> It highly emphasizes common sense reasoning: models are expected to recognize visual elements and then reason the subtle contrast between these elements to identify the oddity.
> For example, for `A lit candle is sitting inside a tightly sealed glass jar` example in the [homepage](https://whoops-benchmark.github.io/), model is expected to realize that *A candle needs a constant supply of oxygen to burn, which does not exist in a sealed bottle, so it is unlikely to see a burning candle inside a sealed bottle*. This benchmark emphasizes common-sense reasoning instead of defying language priors.
>
> ### Qualitative Comparison
> Although its images are creative and out of distribution like ours, this benchmark does not utilize language priors to test the model's susceptibility to linguistic distractions as our benchmark does.
> Especially, in their QA mode(which is the same task as our benchmark), the questions are all very normal questions without any strong language priors. **Some examples can be found at [figure](https://imgdrop.io/image/gRcat)**.
> Also, it uses a format of open-ended questions, which is hard to evaluate and contains ambiguities and even errors.
>
> ### Quantitative Comparison
> Unlike Winoground and our benchmark, Whoops! does not provide control groups or textual components for comparison.
> To measure language priors, we analyze the suggestiveness of questions by evaluating how determined and confident GPT-4o answers them (without visual context).
> A more suggestive question is expected to yield more determined and confident answers, while a less suggestive question is expected to yield more diverse answers.
> For this, we calculate the number of unique answers given by GPT-4o over five attempts while setting its temperature to 1.0 for higher randomness. Semantic differences were normalized to exclude synonyms.
>
> The results reveal:
>
> - Questions from Whoops! yielded **2.58** unique answers (on average) with a standard deviation of **1.48**.
>
> - In our benchmark:
>     - Without facts, GPT-4o gave **1.53** unique answers (std **0.94**).
>     - With facts, GPT-4o gave **1.10** unique answers (std **0.42**).
>
> Interestingly, we find questions from Whoops! yield **2.58** unique answers (in 5 attempts) on average with std **1.48**.
> For our benchmark, without facts, GPT-4o gives 1.53 unique answers (in 5 attempts) on average with std 0.94.
> With facts, GPT-4o gives 1.10 unique answers (in 5 attempts) on average with std 0.42.
>
> While both benchmarks include creative images, these results show that Whoops! questions adhere to conventional styles, which do not prompt GPT-4o to stereotype its answers.
> In contrast, our benchmark's suggestive questions are intentionally designed to elicit stereotype-averse responses, aligning with our design principles.

---

> ### Author Response · Authors · 2024-11-23
> **Author Rebuttal (3/6)**
>
> > Q: I am quite concerned by the quality and content of the generated images. For example, in Figure 3: there is a flamingo present (alongside the tiger) in the image where "tiger" is the correct answer (and "flamingo" is supposed to be
> ). Likewise, the "round" moon is actually boxed in a hexagon. Is the "horse with humps" sufficiently different from a camel? Horses don't have humps: is it fair to say that image is indeed a "horse"? Is the image of Paris with the Red Square really "Paris", or could it be "Moscow" with the Eiffel Tower? And so forth. I have looked at many examples and feel this issue is pervasive in this dataset.
>
> We appreciate the reviewer’s observation regarding the potential distracting elements in our benchmark. These elements are actually intentionally designed to make the benchmark sufficiently challenging for both current and future VLM models.
>
> For instance, in the first image of Figure 3, both a tiger and a flamingo are present and flamingo is explicitly mentioned in the question. However, in the image, the tiger is the only pink animal standing on one leg in the water, perfectly satisfying the question and answer without ambiguity. Despite this intentional distraction, human users can confidently identify the correct answer, achieving over 98% accuracy for those challenging cases, as shown in the “Score” column of Table 3 ($vVLM^F_{score}$, $vVLM^P_{score}$). **This highlights that these distracting elements do not hinder human comprehension.**
>
> We believe that incorporating such elements contributes to the benchmark’s ability to challenge VLM models effectively, making it a valuable and impactful resource for evaluating current and future models.
>
> To address your concern directly, **with no offense, we firmly believe that the quality of the images in our proposed benchmark is high. Moreover, the distracting elements you mentioned are deliberately included to challenge VLMs and assess their ability to handle ambiguity effectively.**
>
> Could you kindly review the additional data samples provided in Appendix A? After considering our rebuttal and the extra data, do you still have concerns about the quality and content of our images? If so, we would greatly appreciate your feedback!

---

> ### Author Response · Authors · 2024-11-23
> **Author Rebuttal (4/6)**
>
> >Q: The "Image DPO" method is not clearly motivated by the topic of "linguistic bias" that is explored in this paper. Why are the perturbations pixel-level and not linguistic? And, I am not sure that this benchmark demonstrates a particular need for more VQA training data. Altogether, the gains (1-3%) from the introduced methods are relatively small compared to the complex system involved (e.g. including image-editing and fine-tuning).
>
>
> The Image DPO approach is motivated by the observation that VLM models, such as Llava, are often misled by textual information while neglecting the visual content, as demonstrated in our benchmark (Table 1). Existing DPO methods encourage models to distinguish between good and bad answers given the same image and question inputs. However, these methods may inadvertently allow models to rely on purely text-based shortcuts by evaluating the quality of the answers alone. In contrast, Image DPO perturbs the image inputs while keeping the questions and answers unchanged. This ensures that VLMs cannot rely solely on textual cues, as the text remains identical across examples. To learn the correct preference, models are trained to focus more on visual information, promoting better integration of multimodal inputs (L288–292).
>
> The Image DPO experiments are not about generating additional data for QA but rather about enabling VLMs to improve themselves by leveraging existing data and tools, without relying on new high-quality data. Current VLM models benefit significantly from training on large amounts of high-quality VQA data. However, this pipeline is not sustainable, as such data is limited and cannot always be readily available. To address this challenge, we approach the problem from a different perspective by enabling VLMs to utilize existing models and seed data to generate more images and QA data and improve themselves.
>
> Furthermore, we enhance Image-DPO performance by scaling the dataset from 80k to 800k samples, **resulting in a notable performance improvement exceeding 1~3%.**
>
> | Model                              | $vVLM^F$ | $vVLM^P$ | NaturalBench-Question | NaturalBench-Image | NaturalBench-Binary | NaturalBench-Group | MMVET | SEED  | $Chair^S$ ⬇️ | $Chair^I$ ⬇️ |
> |------------------------------------|----------|----------|------------------------|--------------------|---------------------|--------------------|-------|-------|--------------|--------------|
> | LLaVA-1.5-7B                      | 29.67    | 37.67    | 37.7                  | 43.8              | 67.3                | 12.7              | 31.1  | 58.6  | 49.1         | 14.8         |
> | LLaVA-1.5-7B + 800k Image-DPO     | 34.17    | 78.67    | 40.68                 | 46.29             | 68.84               | 14.95             | 33.8  | 59.65 | 45.4         | 12.4         |
> | LLaVA-1.5-13B                     | 35.33    | 41       | 39.6                  | 44.6              | 68.9                | 14.8              | 36.1  | 61.9  | 48.3         | 14.1         |
> | LLaVA-1.5-13B + 800k Image-DPO    | 38.14    | 42.5     | 42.68                 | 47.37             | 70.36               | 17.16             | 37.5  | 62.31 | 42.6         | 11.6         |

---

> ### Author Response · Authors · 2024-11-23
> **Author Rebuttal (5/6)**
>
> > Q: Could the authors please elaborate on how they constructed their dataset? In particular, it is not obvious how questions, answers, and images are specifically generated. E.g. what prompts are used for generating questions and images? I believe answers are entirely written by humans: is that correct? These are really important details that are unclear from my reading.
>
> Thank you for your thoughtful question, which allows us to clarify and improve the presentation of our dataset construction process!.We address your concerns below.
>
> Our proposed dataset features question-image-answer (QIA) pairs intentionally designed to challenge state-of-the-art VLMs against language priors. To achieve this, we introduce a heavy-human involved process as it is beyond just prompting existing VLMs.
>
> ## Construction Process:
>
> ### Question-Answer (QA) Generation:
> Parts of the questions and answers are directly authored by humans under the high-level guidance of the design principles listed in Section 3.1
> Some QA candidates are generated using OpenAI-O1 and Claude-3.5-Sonnet with carefully crafted prompts (one example is provided in Appendix D Figure 30). These candidates are then meticulously reviewed and either selected or refined by humans to ensure they meet our criterias.
> ### Image Generation:
> To align with the generated QA pairs, we employ GPT-4 to create diverse and descriptive text prompts for image generation. These prompts are used with advanced image generation models, such as FLUX and DALL-E 3, to produce the corresponding images.
> For each QA pair, multiple images are generated, and human reviewers carefully evaluate them to select the most suitable option or request re-generation as needed. This ensures that the final image is both accurate and fully aligned with the QA context.
> Our prompt used to guide GPT-4 in generating image prompts is also included in Appendix D Figure 31.
> ### Human Review:
> At every stage, the outputs undergo rigorous human review to ensure the quality, clarity, and challenge level of the QIAs. Low-quality, incorrect, or insufficiently challenging QIAs are filtered out during this process.
> Additionally, we test the QIAs with users to confirm they are intuitive for humans to answer while remaining difficult for VLMs.
>
> > Regarding “answers are entirely written by humans: is that correct?”:
>
> Parts of the answers are indeed written by humans, while others are generated by VLMs and subsequently reviewed, refined, or selected by humans to ensure alignment with our dataset’s objectives.
>
> We appreciate your feedback, as it highlights areas where further clarification is valuable. **In response, we have expanded the description of our dataset construction process in the revised paper and included additional details and example prompts in Appendix D to enhance transparency. Do you think it would be beneficial to include an illustration of the annotation process in the main paper?**

---

> ### Author Response · Authors · 2024-11-23
> **Author Rebuttal (6/6)**
>
> > Q: I am not sure that Sec. 3.1 needs to rely on mathematical formulations and the three objectives. I think this actually makes reading the following sections a bit confusing. Could the authors please clarify how these objectives specifically relate to the construction of the questions/answers/images (if at all)? E.g. Sec. 3.2 just says "we designed answers [for/against] language priors" but does not relate this to a particular objective.
>
> Thank you for your suggestions! **We have revised this section (3.1) around both the math and text to improve clarity. Could you please take another look and let us know if it aligns with your understanding or if there's anything still would confuse readers?**
>
> These objectives serve as our guiding principles for constructing the dataset, which we strictly adhere to during all phases of data generation. Specifically, we aim to construct a dataset that satisfies the following properties:
>
> - Text-Only Answer Confidence: Without any accompanying images, the question should naturally lead to a high-confidence $Answer_{\text{prior}}$. This is validated by the results of "GPT-4o (text only)" in Table 1, where it achieves approximately over 90% accuracy without images.
>
> - Clarity with Images: When images are provided, the correct answer should be clearly distinguishable from $Answer_{\text{test}}$ with high confidence, and $Answer_{\text{test}}$ and $Answer_{\text{prior}}$ should be distinct enough to avoid confusion. To achieve this, we carefully generate images based on the question and answers, rigorously verifying the correctness and consistency of each Question-Image-Answer (QIA) pair.
>
> - Language Priors vs. Visual Clarity: Without images, $Answer_{\text{test}}$ should be difficult to infer from the question alone, making it out of the distribution of typical language priors. Conversely, $Answer_{\text{prior}}$ should become incorrect when images are introduced, reducing ambiguity and ensuring the system relies on visual information for accuracy.
>
> We iteratively check, filter and refine the generated question-image-answer pairs to ensure our generated data satisfies the above objectives.

---

> ### Comment · Reviewer_Nay6 · 2024-11-24
>
> **Summary**
>
> Thanks to the authors for their thorough response. I have read their response and checked their revisions. I appreciate the author effort and that they have resolved a few of my concerns. I would now consider my score a "4" (which I am not able to officially select in the review form).
>
> I think this paper would benefit greatly from additional time and revisions. (There are too many proposed revisions and new experiments offered in this short rebuttal period. I think new results need time to sit and be properly be incorporated into the paper.) Moreover:
>
> 1. The writing is simply not clear enough. Details were missing (now provided), but this paper would still benefit from significant restructuring.
>
> 2. It is essential for visual distractors to be removed from the dataset or otherwise controlled in the results.
>
> 3. I do not believe the introduced Image DPO method is strictly motivated by the rest of the paper. I think it is very involved and could be its own paper; it distracts from this one. I recommend the authors replace this with a simpler method (even something naive) that is directly inspired by the "linguistic biases" explored by this paper.
>
> **Details**
>
> **Existing datasets:** The authors have actually satisfied my concern about whether other existing datasets (Winoground and Whoops!) already sufficiently defy language priors. After looking at the Winoground examples shared by the authors, I agree that "defying priors" is not the focus of Winoground (Fig. 1 in Winoground is actually misleading). I would also agree that Whoops! examples defy commonsense priors, not linguistic priors.
>
> **Dataset quality:** That said, I remain concerned about the quality of this dataset. I mentioned that the presence of visual distractors in the proposed dataset is pervasive. (At first, I felt this adds subjectivity to the dataset: but the authors mention that humans score 98% and are able to disambiguate visual distractors.) The authors now (newly) claim that these visual distractors are a deliberate feature and make the benchmark more challenging. However, visual distractors are not addressed or controlled for in the paper; I believe these are actually confounding variables. I don't think we can say to what degree models fail due to visual distractors or the linguistic priors. This clearly indicates a problem with the benchmark design.
>
> **Introduced method:** Regarding Image DPO, the authors now introduced new experiments that train on 10x more samples and improve the performance more significantly than before (1-3%). I accept the other things the authors say, but my primary concern about the relevance of Image DPO remains: "why are the perturbations pixel-level and not linguistic?"
>
> **Writing:** I appreciate the additional clarifications provided by the authors. (1) These dataset construction details need to be included in the main paper. (2) It is still very confusing how the objectives in Sec. 3.1 are specifically used in Sec. 3.2.
>
> I recommend re-writing the details from (1), Sec. 3.1, and Sec. 3.2 as a single section e.g. "Dataset construction":
>
> "We produced a dataset of 300 examples: each consisting of 1 question and 3 image–answer pairs. We first write questions by... We generate images using ... We create the first image–answer pair such that the answer is likely [under this specific definition]. We create the other two pairs such that the answer is unlikely given the question alone [by maximizing other specific objectives]."
>
> This needs to be written in a straightforward way, such that a reader could intuitively / unambiguously create a similar dataset. (Finer details, like hyper-parameters can be deferred to appendix; but relevant details like prompts, how artifacts were generated, and the order artifacts were created in should be obvious.)

---

> ### Author Response · Authors · 2024-11-24
> **Still two key misunderstandings. Also, it's unfair to raise concerns and then dismiss our new experiments as excessive.**
>
> Hi Reviewer Nay6, we thank you for your quick response and the future feedback. However, we would like to kindly point out you still have two fundamental misunderstandings, with no offense. Also, **you have two unfair statements.**
>
>
>
> ---
>
> ## Unfair statement 1
>
> It is both unprofessional and unfair to first raise numerous misunderstandings during the review process, prompting us to address them with new experiments, only to then criticize the rebuttal by stating: *There are too many proposed revisions and new experiments offered in this short rebuttal period. I think new results need time to sit and be properly incorporated into the paper.*
>
> Most of the added experiments directly address your misunderstandings. For instance, your initial misinterpretation of our paper and Winoground + Whoops prompted us to conduct new experiments to clarify these points, which you acknowledged as valid. However, you then criticized the inclusion of these experiments as excessive. It is worth noting that these experiments were not intended to be included in the main paper but were provided solely to resolve your concerns.
>
> As a result, the benchmark data in our submission remains unchanged, and the main paper has only undergone minor modifications.
>
> ---
>
> ## Unfair statement 2
>
> You initially stated that the improvements of our ImageDPO method were limited. **After we addressed this concern and improved the performance, you then suggested to "replace this with a simpler method (even something naive)" + "could be its own paper".** However, our proposed method is directly motivated by our benchmark and observations, and you were unable to provide a clear description of the simpler or naive approach you mentioned.
>
> **If a simple and naive method is worthy here, why our proposed ImageDPO with both theoretical and empirical evidence are meaningless?**
>
> ---
>
> ## Misunderstanding 1
> We think there are some misunderstandings here.
>
> Our submitted version contain both the quantitative and qualitative results regarding "*It is essential for visual distractors to be removed from the dataset or otherwise controlled in the results.*", where we show with exactly the same image and different language priors, how the results would be influenced.  Could you please check Table 1 (now Table 2) and Figure 4 in the submitted paper? This is how we exactly design the metric $vVLM^F$ and $vVLM^P$. The qualitative results show the results for both LLaVA and GPT4O.
>
> **Can you properly acknowledge your statement is false?**: "*authors now (newly) claim that these visual distractors are a deliberate feature.*"
> - Those are the features designed in the benchmark from the beginning, absolute not the new/newly claim for the rebuttal.
>
> ---
>
> ## Misunderstanding 2
>
> **We are confused by the question "why are the perturbations pixel-level and not linguistic?" and "the image dpo is not motivated".  Could you elaborate more about what do you mean by *linguistic perturbation* here? **
>
> - **Image DPO motivation**. Our benchmark reveals that VLM models could overlook visual information and be misled by language priors. So here we explore a method to enforce models to look at visual inputs. The most fundamental intuition of image dpo is: within the preference pair, we keep all the text (question, answer) the same, while only modifying the images. In this way, model has to look at the input images to decide which answer is preferred or not. We iteratively explain this  part in the submitted paper and rebuttal. Could you elaborate more why you think image dpo is not motivated?
>
> - If your *linguistic perturbation* means semantically perturbs the images,   we **did** include linguistic perturbations to images, which is "semantic corruption" shown in Figure 2. We adopt semantic changes to images and make semantic-level perturbation.
>
> - If your *linguistic perturbation* means why not changing the text of the QAs, we believe our motivations  and experimental results explain the underlying reasons.
>
>
> ---
>
> We are a bit confused, as we initially understood your comments as an effort to provide constructive feedback for improving the paper, given the detailed points you raised. However, after addressing these concerns and revising the paper, we notice that new issues have been raised and **most of which are beyond the research scope, while others seem to reflect misunderstandings about aspects already clarified in the submitted version.**

---

> > ### Comment · Reviewer_Nay6 · 2024-11-25
> >
> > I cannot engage much further in this discussion, as I have already put my best-faith efforts into reviewing this work, but my "professionalism" and "fairness" are being called into questioned. I apologize for any misunderstandings in or offense caused by my reviews. My intention is to be constructive and a diligent reviewer.
> >
> > I would clarify that:
> >
> > 1. *Misunderstanding 1*: I am referring to the visual distractors in the images, not the distractor facts that are part of the benchmark's evaluation.
> >
> > 2. *Misunderstanding 2*: The "semantic corruption" in this paper is underspecified. The only definition I can find in the paper is in Appx. E.1: "For semantic corruption, we use the pretrained Instruct-Pix2Pix model to modify a single object in the image while keeping the rest unchanged." (L1355-1356) What are the prompts? Which object is modified and how? I do not understand what this corruption is and whether it corresponds with a "linguistic" perturbation as I have been asking about.
> >
> > 3. *Unfair Statement 1*: **I did not ask or even imply that the authors should conduct more experiments in my review.**
> >     - I mentioned that "Winoground and Whoops! are vision–language benchmarks that [also focus on defying language priors]". I was mistaken; this is a harmless misunderstanding: the authors sufficiently corrected me with an explanation and by showing qualitative examples from both datasets. In my view, there was no need for the authors to run quantitative experiments to compare against these other datasets.
> >     - I also pointed out that the originally reported gains from the Image DPO method were small. Of their own volition, the authors tried to address this concern by running more experiments and training on 10x as much data.
> >
> > 4. *Unfair Statement 2*: My claims about Image DPO have always been that: Image DPO is a very complex method, the initial reported gains were too small in comparison, and that I don't see how it is inspired by the topic of "linguistic bias" that is the focus of this paper. The authors improved the gains, but the last point is still unresolved as of (2) above. When I now suggest that "Image DPO could be its own paper", I am referring to the complexity (now even more obvious by the revised Appx. B). When I suggest a "simpler method directly inspired by linguistic biases", for example: fine-tuning is simpler than DPO; the generated data could just focus on these linguistic biases (rather than the pixel-level corruptions that I am indicating are irrelevant).
> >
> > The authors insist that I have misunderstandings. To give them the benefit of the doubt, I will reduce my "confidence" from 4 to 3. I admit one misunderstanding about related work, which was minor and easily resolved. I read the paper multiple times now. If I do continue to have misunderstandings, I must defer these to the lack of clarity in the text, which I have consistently commented about. I maintain my belief that this paper needs more revisions before it is ready for publication. I hope the authors will use my feedback in said revisions.

---

> > > ### Comment · Reviewer_Nay6 · 2024-11-25
> > >
> > > I must also note that the most recent comment from the authors has been edited 16 times, including 3 edits *after* my response was posted. The current version has significant differences from what I responded to. I think I was responding to the edit at Nov 24, 10:30 AoE.

---

> > > > ### Author Response · Authors · 2024-11-25
> > > > **We update the comment for better presentation**
> > > >
> > > > Thank you for clarifying this. We updated the comment to better convey our thoughts. FYI, we also update the structure and some words in our rebuttal for better presentation
> > > >
> > > > The three edits made after your response were merely formatting changes, such as bolding, which you can easily verify. Could you please verify this point and update your conclusion here in your comment?

---

> ### Author Response · Authors · 2024-11-25
> **Thanks, our thoughts, and you've missed one point**
>
> We sincerely thank you for your review, Reviewer Nay6. You are a constructive and a diligent reviewer. This is the reason why we have approached the rebuttal with the utmost seriousness, resulting in detailed response. Before you close the discussion, we would like to share our feelings, highlight one detail you missed, and the simple baseline is to expect failure.
>
> ---
>
> >  What are the prompts? Which object is modified and how?
>
> We included the prompt details and example in the submitted version Figure 13 & Figure 10 (now Figure 14 & 11). Which object is modified and how is defined by the VLM itself. We are happy to include more details if you feel responsible and willing to engage more.
>
>
>
> ---
>
> **We are frustrated because your response to our rebuttal gave us a feeling that you have a fixed "perfect" vision of how our paper should be, dismissing much of our current organization, contributions, and value. This response comes while we are still in the middle of the rebuttal process and have already addressed most of your initial concerns.**
>
>
> You new concerns are high-level, one-line one-sentence, one "non-solvable" fundamental issues, such as too many proposed revisions, image-dpo is too complex, why are the perturbations pixel-level and not linguistic?. Regarding the last point, only after you elaborated on your thoughts did we realize you had missed the details we outlined in the appendix. Also, we are not unresponsive, while we are happy to discuss them with you more.
>
>
> Your initial reviews raised detailed concerns, which we addressed with considerable effort and creativity, conducting appropriate experiments to resolve them thoroughly within the short time. Once these detailed concerns were addressed, you raised a different issue, stating, *There are too many proposed revisions and new experiments offered in this short rebuttal period. I think new results need time to sit and be properly incorporated into the paper*.
>
>
> The experiments we conducted were genuinely aimed at addressing your concerns comprehensively, providing both qualitative and quantitative results. However, it is challenging for us to anticipate your perspective, such as *In my view, there was no need for the authors to run quantitative experiments to compare against these other datasets,* without your direct feedback during the process. Also, many of them will not be incorporated into the main paper, but listing in the appendix, still you insisted those are too much for revision. Furthermore, ICLR encourages our authors could update the manuscript, and we update the main paper actually not a lot.
>
>
> We hope you can think our communications from the author's perspective and understand our frustration.
>
> ---
>
>
>
> > fine-tuning is simpler than DPO; the generated data could just focus on these linguistic biases (rather than the pixel-level corruptions that I am indicating are irrelevant).
>
> Our proposed ImageDPO is grounded in existing research and not developed in isolation. Several prior works (Zhou et al., 2024a; Deng et al., 2024; Zhou et al., 2024c; Wang et al., 2024b;a), as cited in our related work, **have demonstrated that simple fine-tuning on synthetic data leads to performance degradation in VLMs**. Furthermore, Table 3 in our paper provides evidence that the proposed ImageDPO achieves better performance compared to these prior DPO methods.
>
>
> ---
>
> **Please select your confidence and scores based on your best judgment. We kindly ask you not considering "To give them the benefit of the doubt".** We, one of the thousands submissions, are making the standard for the review process in AI/ML community.

---

### Official Review · Reviewer_Dj1v · 2024-11-06

**Soundness:** 3
**Presentation:** 4
**Contribution:** 4
**Rating:** 8
**Confidence:** 4

**Summary:**

This paper introduces vVLM, a benchmark to address the issues of language biases in existing VQA benchmarks. vVLM consists of 300 questions, where each question is accompanied by a distractor fact and three image-answer pairs, where only one pair agrees with the distractor fact and the other two contradict the distractor. The intuition is that a VLM that is not guided by language priors should be able to ignore the distractor fact and answer. The authors show that existing SOTA VLMs such as GPT-4o struggle with this benchmark. To improve the performance of VLMs, the authors also propose a DPO-based method that self-generates VQA pairs for training using image generation and editing models, and then creates dispreferred pairs for DPO training by introducing pixel-level and semantic corruptions in the images.

**Strengths:**

- Impressive data collection and curation effort for the benchmark, with multiple iterations of human filtering and human-AI collaboration. Manual creation of synonym set for the answers is a good
- Evaluated VLMs on both OOD and prior subsets separately, both with and without the distractor facts
- Good baselines for ImageDPO method, and also evaluated on multiple datasets

**Weaknesses:**

- **The Image-DPO method is weakly motivated:** DPO is used to identify which is the preferred response out of two different responses for the same input, but the proposed method does the opposite — says which is the preferred input for the same output? Model is being trained to learn what is the preferred input, what is the intuition connecting this to the proposed goal (improving visual reasoning)?
    - repeat Text-DPO baseline, but with the generated images used for DPO as images for additional image-question-answer triplets (i.e. if you had a pair of triplets (Q, I, A_w) and (Q, I, A_l) for TextDPO, then for your perturbed images I’ with the blurring and pixelation perturbations, create a new triplet pair (Q, I’, A_w) and (Q, I’, A_l) ). My suspicion is that training with perturbed images at all is what is providing the additional performance benefit for your method, rather than the specific DPO objective you’re using.

- **Missing multiple seeds and significance testing:** The numbers in Tables 2 and 3 are the result of a single training run. Reporting the mean and standard deviation of 3 training runs would be much more informative about how reliable these results are.

- Language biases from internet-scale data (used to motivate the problem) is different from how language bias is operationalized here (distractor text included into the model context)

**Questions:**

- Eq2 should have I as a conditioning variable, π(a_test | Q, I) ≥ δ
- Unclear how the alternate answers are selected; are they manually selected by the people designing the dataset?
- the acronym for VLM is expanded twice in L170
- What does vVLM stand for? It's never mentioned
- For the conclusion that images that were more challenging to generate were more difficult to answer for the VLMs (L464-469) — are there any numbers/tables pointing to the same? How is “easily generated” evaluated?
- Wrong citation for MMEC in L527: I think you meant this paper https://arxiv.org/pdf/2306.13394

---

> ### Author Response · Authors · 2024-11-23
> **Author Rebuttal (1/4)**
>
> We sincerely appreciate the insightful and detailed feedback from Reviewer Dj1v! They really help improve our paper, thank you! We seriously address your comments in the below and have integrated many of them into the revision. We also greatly appreciate your recognition of our efforts in curating the benchmark and ensuring its quality and accessibility for effective evaluation.
>
> ---
>
> > Q: DPO is used to identify which is the preferred response out of two different responses for the same input, but the proposed method does the opposite — says which is the preferred input for the same output? Model is being trained to learn what is the preferred input, what is the intuition connecting this to the proposed goal (improving visual reasoning)?
>
> > repeat Text-DPO baseline, but with the generated images used for DPO as images for additional image-question-answer triplets (i.e. if you had a pair of triplets (Q, I, A_w) and (Q, I, A_l) for TextDPO, then for your perturbed images I’ with the blurring and pixelation perturbations, create a new triplet pair (Q, I’, A_w) and (Q, I’, A_l) ). My suspicion is that training with perturbed images at all is what is providing the additional performance benefit for your method, rather than the specific DPO objective you’re using.
>
> This one is really a fundamental point. In the below, we first describe how our objective is actually optimizing the upper bound of the RLHF objective (detailed math is included in Appendix B) and then we demonstrate the intuition of why our objective could help model focusing on visual inputs. Furthermore, We conducted a verification experiment to demonstrate that the TextDPO objective, when applied with perturbed images and varied answers, does not significantly enhance performance on several benchmarks compared to the ImageDPO objective.
>
> ## How our ImageDPO objective is actually the upper bound of the RLHF objective
>
> In the below, we mathematically describe the connection between our objective and RLHF by constructing preferred pairs $p((Q, I_{w}, A) \succ (Q, I_{l}, A))$. Here, $I_{w}$ represents the original image, while $I_{l}$ denotes a corrupted image generated through semantic corruption, Gaussian blur, or pixelation. $Q$ and $A$ stand for question and answer, respectively. Please refer to our updated Appendix B for more thorough proof.
>
> Similarly, we adopt the BT model to represent the joint preference distribution: $p((Q, I_{1}, A) \succ (Q, I_{2}, A)) = \frac{\exp\left( rQ, I_w, A)\right)}{ \exp \left( r(Q, I_w, A)\right) + \exp \left( r(Q, I_l, A) \right)}$
>
> Also, according to the DPO paper (Rafailov et al. 2023), a straightforward optimal solution to the KL-constrainted reward function maximization object in
> $$ L_R(r_{\phi}, S) = -E_{(A,Q, I_w, I_l) \sim S} [ \log \sigma(r_{\phi}(Q, I_w, A) - r_{\phi}(Q, I_l, A)) ] $$
>  is: $\pi_{r}(A | Q, I) = \frac{1}{ Z(Q, I)}\pi_{\text{ref}}(A | Q, I)\exp(\frac{1}{\beta}r(Q, I, A))$, where $Z(Q,I) = \sum_{A}\pi_{\text{ref}}(A|Q, I)\exp(\frac{1}{\beta}r(Q, I, A)$ is a partition function. Here $r$ should be any reward function, which makes $Z$ hard to tract.
>
>
> We plug the above optimal solution into the BT model, we have:
>
>
> $$
>         p((Q, I_{1}, A) \succ (Q, I_{2}, A)) =
>     \frac{\exp \left( \beta \log \frac{\pi(A|Q, I_w)}{\pi_{\text{ref}}(A|Q, I_w)
>     } + \beta \log Z(Q, I_w) \right)}{\left( \beta \log \frac{\pi(A|Q, I_w)}{\pi_{\text{ref}}(A|Q, I_w)
>     } + \beta \log Z(Q, I_w) \right) + \left( \beta \log \frac{\pi(A|Q, I_l)}{\pi_{\text{ref}}(A|Q, I_l)
>     } + \beta \log Z(Q, I_l) \right) }
> $$
> $$=\frac{1}{1 + \exp(\beta \log\frac{\pi(A|I_l, Q)}{\pi_{\text{ref}}(A|I_l, Q)} - \beta \log\frac{\pi(A|I_w, Q)}{\pi_{\text{ref}}(A|I_w, Q)} + \beta \log Z(I_l, Q) - \beta\log Z(I_w, Q))}
> $$
> $$
> = \sigma \left( \exp(\beta \log\frac{\pi(A|I_l, Q)}{\pi_{\text{ref}}(A|I_l, Q)} - \beta \log\frac{\pi(A|I_w, Q)}{\pi_{\text{ref}}(A|I_w, Q)} + \beta \log Z(I_l, Q) - \beta\log Z(I_w, Q)) \right)
> $$
>
> Now that we have the probability of human preference data in terms of the optimal policy rather than the reward model, we can formulate a maximum likelihood objective for a policy $\pi_{\theta}$. Our policy objective is :
>
> $$ L(\pi_{\theta};\pi_{ref}) = E_{(Q, A, I_{w}, I_{l}) \sim S} \Bigg[ -\log \sigma \Bigg( \beta \log\frac{\pi_{\theta}(A|I_w, Q)}{\pi_{ref}(A|I_w, Q)} - \beta \log\frac{\pi_{\theta}(A|I_l, Q)}{\pi_{ref}(A|I_l, Q)} + \beta \log Z(I_w, Q) - \beta\log Z(I_l, Q) \Bigg) \Bigg] $$
>
> You can observe the distinction between our objective and the DPO objective. In the DPO objective, since the different outputs while the same input are involved, the subsequent $ \log Z $ term can be eliminated. However, in our objective, this term remains present as we condition on the inputs. Accordingly, we follow the next-comment steps to eliminate the $Z$ term, as its structure is challenging to track analytically.

---

> ### Author Response · Authors · 2024-11-23
> **Author Rebuttal (2/4)**
>
> As $f(x) = -\log \sigma(x)$ is a convex function ($\sigma$ is the sigmoid function), we can apply Jensen's inequality $f(\frac{1}{2} x + \frac{1}{2}y) \leq \frac{1}{2}f(x) + \frac{1}{2}f(y)$:
> $$
>      L(\pi_{\theta};\pi_{\text{ref}}) \leq E \Bigg[ -  \frac{1}{2} \log \sigma\Bigg( 2\beta \log\frac{\pi_{\theta}(A|I_w, Q)}{\pi_{\text{ref}}(A|I_w, Q)} - 2\beta \log\frac{\pi_{\theta}(A|I_l, Q)}{\pi_{\text{ref}}(A|I_l, Q)} \Bigg) -  \frac{1}{2}\log \sigma \Bigg( 2\beta \log Z (I_w, Q) - 2\beta\log Z (I_l, Q) \Bigg) \Bigg]
> $$
>
>
> As $\log \sigma \left( Z(I, Q) \right)$ is not a function of $\pi_{\theta}$, the above objective is equivalent to the below Equation, where $\alpha = 2\beta$. It is the same as our final objective, also listed in the Equation (4) of our main paper.
>
> $$ L(\pi_{\theta};\pi_{\text{ref}}) \leq  -E_{(Q, I_w, I_l, A) \sim S} \Bigg[ \log \sigma \Bigg( \alpha \frac{\pi_{\theta}(A \mid Q, I_w)}{\pi_{\theta_{\text{ref}}}(A \mid Q, I_w)} - \alpha \frac{\pi_{\theta}(A \mid Q, I_l)}{\pi_{\theta_{\text{ref}}}(A \mid Q, I_l)} \Bigg)\Bigg] $$
>
> ## Our intuition
>
> Our goal is to strengthen the model’s dependence on visual inputs, minimizing the dominance of text input, to enhance its visual reasoning capabilities. Please refer to this [image](https://imgdrop.io/image/grad-diff.gZPlT), where we illustrate the gradient flow comparisons between ImageDPO and TextDPO.
>
> In the TextDPO approach, the model receives positive gradients (green arrows) for the preferred answer, denoted as $A_w$, and negative gradients (red arrows) for the dispreferred answer, $A_l$. Combining these gradients directly encourages the model to produce the desirable answer while minimizing the likelihood of generating the dispreferred one.
>
> Conversely, our proposed ImageDPO approach applies positive gradients when the preferred image $I_w$ is provided as input and negative gradients for the dispreferred image $I_l$, while maintaining the same output answer. Since both approaches share identical outputs and input text, but with opposite gradients due to the inclusion of the original and corrupted images, the model is compelled to leverage its visual input branch to achieve the gradient reversal. This mechanism pushes the model to focus more on visual inputs. When combined with a supervised fine-tuning (SFT) objective, this approach further reinforces the model’s attention to visual inputs, resulting in improved performance in scenarios where visual input plays a crucial role.
>
>
> ## Your suggested experiments
> We conducted the suggested experiment, i.e., using corrupted images and different preferred answers $(Q, I’, A_w)$ and $(Q, I’, A_l)$. The results below demonstrate that Image-DPO consistently outperforms its Text-DPO counterparts across various metrics.
>
> | Model                                       | $vVLM^F$ | $vVLM^P$ | MMVET | SEED  | $Chair^S$ ⬇️ | $Chair^I$ ⬇️ |
> |---------------------------------------------|----------|----------|-------|-------|--------------|--------------|
> | LLaVA-1.5-7B                                | 29.67    | 37.67    | 31.1  | 58.6  | 49.1         | 14.8         |
> | LLaVA-1.5-13B + Text-DPO-over corrupted-Images | 31       | 37.5     | 30.3  | 59.19 | 48.2         | 13.5         |
> | LLaVA-1.5-7B + Image-DPO-SameScale          | 32.83    | 38.33    | 32.8  | 59.3  | 47.2         | 13.1         |

---

> ### Author Response · Authors · 2024-11-23
> **Author Rebuttal (3/4)**
>
> > Q: Missing multiple seeds and significance testing: The numbers in Tables 2 and 3 are the result of a single training run. Reporting the mean and standard deviation of 3 training runs would be much more informative about how reliable these results are.
>
> We thank you for this valuable suggestion! **We’ve incorporated the first round of llava-13b experiments as shown in the below table.** We also adopted NaturalBench (Li et al. 2024), a benchmark recommended by Reviewer M3UT—that evaluates performance against natural adversarial samples, focusing on aspects such as compositionality and biases. The results indicate that our proposed image-dpo is reliable with small variance for different running.
>
> | Model                          | $vVLM^F$         | $vVLM^P$         | NaturalBench-Question | NaturalBench-Image | NaturalBench-Group | MMVET           | SEED            | $Chair^S$ ⬇️      | $Chair^I$ ⬇️      |
> |--------------------------------|------------------|------------------|-------------------|----------------|----------------|-----------------|-----------------|----------------|----------------|
> | LLaVA-1.5-13B                 | 35.33           | 41               | 39.6             | 44.6           | 14.8           | 36.1            | 61.9            | 48.3           | 14.1           |
> | LLaVA-1.5-13B + Image-DPO     | 38.10 $\pm$ 0.09 | 42.39 $\pm$ 0.10 | 42.62 $\pm$ 0.08 | 47.27 $\pm$ 0.11 | 17.07 $\pm$ 0.30 | 37.13 $\pm$ 0.72 | 62.32 $\pm$ 0.01 | 42.63 $\pm$ 0.15 | 11.53 $\pm$ 0.12 |
>
>
> Due to computation constraints, we are still running experiments to report  the mean and standard deviation of 3 training runs for all of our models, including cambrian-8B and llava-7B. **We expect to have results for cambrian-8B before the end of discussion and for llava-7B later. Once we have the complete results, we will then update the results in the revision.** Thank you for the suggestion!
>
>
> ---
> > Q: Language biases from internet-scale data (used to motivate the problem) is different from how language bias is operationalized here (distractor text included into the model context)
>
> This is an excellent point! Below, we present quantitative experiments to demonstrate that the language biases in our benchmark are, to some extent, aligned with those found in internet-scale datasets. Additionally, we argue that these biases are not confined to the distractor facts but are inherent throughout the dataset.
>
> To investigate this, we leverage GPT-4o, a model trained on internet-scale data, which we hypothesize to inherently reflect the linguistic biases present in such datasets. Specifically, we prompt GPT-4o to answer our questions without access to images, repeating the process five times for each question to collect five diverse responses. To enhance diversity, we set the temperature to 1.0, introducing randomness in the generated answers. We then calculate the probability of GPT-4o’s responses matching the answers from our benchmark. Note that, for this experiment, we eliminate image-related phrases from the prompt during this text-only inference, such as removing expressions like “as shown in the image".
>
> The results indicate that our prior answer, Answer 1, aligns with GPT-4o’s predictions **94.67%** of the time, while our out-of-distribution answers, Answers 2 and 3, occur with probabilities of **1%** and **0.67%**, respectively.
>
> We further repeated the experiment after removing the distractor text from the questions. In this setup, the prior answer still matches with GPT-4o’s predictions **77%** of the time, while Answers 2 and 3 occur with probabilities of **0.67%** and **0.33%**, respectively.
>
> These findings demonstrate that our questions and answers reflect language biases consistent with those present in internet-scale data to some levels, even without the explicit inclusion of distractor facts. Do you think this experiment is set up well? Please let us know if you have any further suggestions!

---

> ### Author Response · Authors · 2024-11-23
> **Author Rebuttal (4/4)**
>
> > Q: Unclear how the alternate answers are selected; are they manually selected by the people designing the dataset?
>
> Yes, the alternate answers are manually designed by humans, as detailed in the second paragraph of Section 3.2. During the dataset creation process, human designers collaborate with OpenAI’s o1 and sonnet-3.5 models. While some answers are suggested by these models, the majority are crafted directly by humans. This approach ensures the benchmark poses significant challenges, making it difficult for commercial VLMs, and serves as a meaningful benchmark for nowadays VLMs.
>
> ---
> > Q: What does vVLM stand for? It's never mentioned
>
> We initially aimed to align our project’s name with the concept of ‘Making the V in VQA Matter: Elevating the Role of Image Understanding in Visual Question Answering,’ as our focus is on enhancing the visual aspect within VLMs. However, we are considering switching to a new name centered around the acronym LAP, which stands for LAnguage Prior. We would love to hear your thoughts on this change and would greatly appreciate any suggestions or feedback you may have!
>
> ---
> > Q: For the conclusion that images that were more challenging to generate were more difficult to answer for the VLMs (L464-469) — are there any numbers/tables pointing to the same? How is “easily generated” evaluated?
>
> We are confident in our conclusion that images that were more challenging to generate proved more difficult for the VLMs to answer. Unfortunately, we did not retain quantitative records specifically tracking generation metrics for each image. However, we did present qualitative analyses during the rebuttal phase that highlight the correlation between image generation complexity and VLM performance.
> **Please refer to [image](https://imgdrop.io/image/gcf3I) for qualitative results**, where we provide multiple image pairs for analysis. In each pair, the left-hand image was easy to generate because it corresponded to our prior answers which align with the language priors. Similarly, the right image in the first row was also straightforward to generate. For instance, tasks like changing the color of a bridge to yellow, adding wings to a sloth, or making a koala jump required only one or two trials to achieve satisfactory results.
>
> Conversely, the right images in the second row illustrate more complex scenarios. For example:
>
> - In the second row, second column, generating an image of a man riding a seal was straightforward, but incorporating additional elements—such as placing the scene on a yellow grass field and adding horses in the background—proved significantly more challenging.
> - In the second row, fourth column, generating an image of a person on a raft was easy, but creating a gondola nearby and ensuring it was empty was far more difficult.
> - In the second row, sixth column, generating a honeycomb with cube/square shapes posed a unique challenge, as honeycombs are predominantly hexagonal. (Note: both cube/square and hexagon/hexagonal answers were accepted as correct in our evaluation pipeline.)
>
> Those hard-to-generate images often required over 50 iterations to produce a satisfactory image. As a general observation, VLMs performed well on images that were easy to generate, even when using weaker models like LLaVA-7B. However, their performance consistently degraded with those requiring significant iterative refinement. This limitation was observed across both open-source models and commercial ones, such as GPT-4O and Claude-Sonnet-3.5.
>
>
>
> ---
>
> > Q: Eq2 should have I as a conditioning variable, π(a_test | Q, I) ≥ δ
>
> > Q: the acronym for VLM is expanded twice in L170
>
> > Q: Wrong citation for MMEC in L527: I think you meant this paper https://arxiv.org/pdf/2306.13394
>
> Thank you very much for bringing them up! We’ve corrected them in the revision.

---

> > ### Comment · Reviewer_Dj1v · 2024-11-25
> > **Response to Rebuttal**
> >
> > Thank you for the detailed response. I am keeping my score the same.

---

### Author Response · Authors · 2024-11-23
**Author response to all reviewers and AC**

We would like to sincerely thank all the reviewers for their high-quality reviews. These reviews stand out as some of the most thoughtful and detailed the first author has received in a long time. As a professional courtesy, we have approached the rebuttal with the utmost seriousness and respect. While our responses are detailed, they include well-designed experiments and thoughtful rebuttals that address each concern comprehensively.

**To summarize the updates made during the rebuttal phase**:

- We compared our proposed benchmark with several other datasets, including Winoground (Thrush 2022 et al.), Whoop! (Bitton-Guetta 2023 et al.), and HallusionBench (Guan 2024 et al.).
    - These comparisons are used to address potential reviewer misunderstandings and will be included in the appendix. We had related discussion in the related works.
- We provided a theoretical proof demonstrating that our ImageDPO objective optimizes the upper bound of the RLHF objective. This encourages the model to effectively utilize its visual input branch by achieving gradient reversal when presented with different input images, while keeping the same input text and output answer [illustration figure](https://imgdrop.io/image/grad-diff.gZPlT).
    - This proof is also included in the appendix, as it does not affect the objectives or experimental results presented in the paper.
- We improved ImageDPO performance by scaling the data.
    - The updated results are reflected in the tables in the main paper, but the overall pipeline remains unchanged. The only modification was the addition of more data generated by the same proposed pipeline.

Most updates during the rebuttal focused on clarifying misunderstandings and demonstrating improved ImageDPO performance. These revisions enhance the submission without altering its core contributions or the proposed benchmark dataset. All contributions remain consistent with the original submission, and for transparency, revised sections in the main paper are highlighted in red (we did not highlight some minor changes).

We kindly seek the AC’s guidance in assessing whether our revisions constitute significant changes to the paper.

---

### Comment · Reviewer_Nay6 · 2024-12-03
**Summary of reviewing thread [Nay6]**

The thread under my review became quite long, so I will do my best to summarize the resolutions, my current viewpoints, and advice for the authors. I have read the full review thread and checked the main paper and appendices, once again. I will only address the *productive* discourse, i.e. that which helps further the evaluation of this work. I am not re-iterating "strengths" which can be seen in my original posting.

**Writing:** I believe the paper is in need of significant textual revisions. I appreciate the extensive additions to the appendices by the authors in revision. However, I believe the main paper needs re-writing; it should clearly cover all high-level details (whereas the appendices are for implementation-level details). One example is what a "semantic corruption" is — nearly all details are in the appendix, yet this is an important part of the proposed method and is even mentioned in the abstract. There was lots of back-and-forth in the reviewing thread about this. As another example, I sketched an outline (at the bottom of [this comment](https://openreview.net/forum?id=lCqNxBGPp5&noteId=xDoouAAKS6)) for what a clearer "dataset construction" section could look like. More examples of what is unclear may be found in the thread.

**Weaknesses:** The authors addressed Weakness (1) that I raised. In my view, the remaining weaknesses [(2) data quality and (3) relevance of the proposed method] are still not clearly resolved.

**(2) Data quality:** I mentioned the pervasive presence of visual distractors in my original review. One example is the "round" moon that is boxed in a hexagon (Figure 1). [In retrospect, I am realizing the flamingo example I originally highlighted may actually be fine. But I gave three more examples and continued to see this pattern through the data.] The authors pointed out that humans score well regardless of the distractors, which is good. However, the authors insist that *visual* distractors are part of the design of this dataset and are controlled in their studies. I really disagree with this claim. In the paper, I see many instances of *textual* distractors (i.e. "distractor facts") and evaluations of these settings (vVLM$^F$ and vVLM$^P$). As far as I can see, there is no commentary whatsoever about visual distractors (the writing should be updated if so). I do not see these being controlled for and I think these are significant confounders to the benchmark. This is actually a really fundamental issue and is the primary reason I consider this work less "sound".

**(3) ImageDPO:** I was concerned with the proposed method because of (a) complexity and (b) relevance. (a) In my original opinion, the method was overly complex relative to rather small gains. The authors scaled their experiment and the gains became larger, so this became a stylistic criticism from my point of view. (b) The research questions of this paper are about the linguistic biases of vision–language models. I was concerned that the proposed method is not motivated by this topic (and is more like a method that does well on the proposed benchmark). The authors insist that the "semantic corruptions" capture this. However, as above, I believe these are very underspecified in the main paper. And, even after revisiting appendices, I continue to find this unintuitive (now I have more (but not all) implementation-level details, like the prompts and the model used and no high-level details, about what semantic corruption is really supposed to do). I think the writing needs to be resolved, before this point can be revisited.

In all, I continue to vote to reject this paper for the above reasons, even after substantive discourse with the authors. I hope the authors will find my advice helpful in longer-term revisions to this work.

---

> ### Author Response · Authors · 2024-12-03
> **Author response**
>
> We greatly thank reviewer Nay6 for their detailed comments and valuable efforts!
>
>
> Beside, we kindly request the AC's attention to [this thread](https://openreview.net/forum?id=lCqNxBGPp5&noteId=0NwPR6izsP), as some arguments raised by Reviewer Nay6 extend beyond the intended scope of our research. To facilitate your review, we have outlined the potential points of contention within the thread.

---

### Meta-Review · Area_Chair_fbcP · 2024-12-23

**Metareview:**

This paper introduces vVLM, a benchmark designed to evaluate vision-language models' (VLMs) reliance on language priors versus visual reasoning, along with a proposed ImageDPO training method to improve visual reasoning capabilities. The benchmark consists of 300 questions paired with three image-answer combinations each - one that aligns with language priors and two that require careful visual analysis.

Initial reviewer opinions were mixed, with concerns raised about the relationship to existing benchmarks like Winoground, the quality and potential visual distractors in the dataset images, and the motivation and relatively small gains from the proposed ImageDPO method. Through the rebuttal period, the authors effectively addressed several concerns by demonstrating through quantitative analysis how their benchmark differs from existing ones in testing language priors and clarifying that visual complexity is an intentional design choice validated by high human performance.

However, some reviewers remained concerned about the writing clarity, particularly around dataset construction details and objectives. There were also ongoing discussions about whether visual distractors should be controlled for in the evaluation. While the authors provided thorough responses and additional experiments, reviewers suggested the paper would benefit from significant revisions to improve clarity and better integrate the proposed method. The final consensus appears to lean negative, with reviewers acknowledging the interesting benchmark contribution but expressing reservations about presentation and methodological aspects.

**Additional Comments On Reviewer Discussion:**

See above.

---

### Decision · Program_Chairs · 2025-01-22

Reject